

# New physics through flavor tagging at FCC-ee

Admir Greljo⋆, Hector Tiblom† and Alessandro Valenti‡

Department of Physics, University of Basel,
Klingelbergstrasse 82, CH-4056 Basel, Switzerland

⋆ admir.greljo@unibas.ch , † hector.tiblom@unibas.ch , ‡ alessandro.valenti@unibas.ch

## Abstract

Leveraging recent advancements in machine learning-based flavor tagging, we develop an optimal analysis for measuring the hadronic cross-section ratios $R_b$, $R_c$, and $R_s$ at the FCC-ee during its $WW$, $Zh$, and $t\bar{t}$ runs. Our results indicate up to a two-order-of-magnitude improvement in precision, providing an unprecedented test of the SM. Using these observables, along with $R_\ell$ and $R_t$, we project sensitivity to flavor non-universal four-fermion (4F) interactions within the SMEFT, contributing both at the tree-level and through the renormalization group (RG). We highlight a subtle complementarity with RG-induced effects at the FCC-ee's $Z$-pole. Our analysis demonstrates significant improvements over the current LEP-II and LHC bounds in probing flavor-conserving 4F operators involving heavy quark flavors and all lepton flavors. As an application, we explore simplified models addressing current $B$-meson anomalies, demonstrating that FCC-ee can effectively probe the relevant parameter space. Finally, we design optimized search strategies for quark flavor-violating 4F interactions.

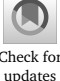

# 1 Introduction

The Future Circular $e^+e^-$ Collider (FCC-ee) [1], a next-generation collider that could one day encircle CERN, promises to open a new frontier in particle physics. With its unprecedented luminosity, the FCC-ee offers exceptional sensitivity to study the electroweak (EW) scale, rigorously testing the Standard Model (SM) while searching for subtle signals of new physics (NP).

The FCC-ee emerges as the envisioned next stride in high-energy physics, building on the foundational achievements of the Large Electron–Positron Collider (LEP) and the Large Hadron Collider (LHC). Through its precise EW measurements, LEP indirectly probed the multi-TeV energy scale, ruling out sizable portions of Beyond the SM (BSM) theories even before the LHC began direct searches [2]. It also provided an in-depth examination of the EW scale, where one-loop quantum corrections served as essential tests of the EW theory. Similarly, the FCC-ee, with its unprecedented statistical power, aims to extend indirect exploration into the multi-10 TeV range—an order of magnitude beyond LEP—and potentially anticipate direct discoveries at the succeeding hadron collider, FCC-hh [3]. Additionally, it will conduct a thorough indirect examination of the TeV scale, shedding light on corners that remained obscure even after the LHC's direct searches. This remarkable sensitivity to generic BSM models at the TeV scale stems from quantum corrections that subtly influence EW precision observables, as shown in [4–7] and further elaborated here. Ultimately, the FCC-ee will establish a comprehensive consolidation of the EW scale, enabling precise tests of two-loop electroweak corrections and advancing our understanding of the SM dynamics.

This paper focuses on the search for new short-distance physics that could manifest through four-fermion (4F) contact interactions. We approach this by working within the framework of the Standard Model Effective Field Theory (SMEFT) at the dimension-six level, employing the Warsaw basis [8] and concentrating on $4q$, $2q2\ell$, and $4\ell$ operators with all possible chirality structures. For a comprehensive compilation of current limits and global fits, see [9, 10], and for related projection studies at future colliders, see [11–14]. These operators are, for instance, generated at the tree-level by integrating out a heavy bosonic mediator above the EW scale, which couples linearly to the SM fermion bilinears [15].

The parameter space of dimension-six 4F operators is vast, primarily due to different flavors [16, 17]. In this work, particular attention is given to non-universal flavor-conserving

interactions. While flavor-changing neutral currents are already tightly constrained, the exploration of the TeV scale demands further scrutiny of flavor-conserving interactions. The LHC has already probed universal NP at the multi-TeV scale; however, non-universal contact interactions, especially those involving heavy flavors, remain poorly constrained [18]. Notably, sizable third-family contact interactions are motivated by theories that address the stability of the electroweak scale. Contact interactions that mimic the hierarchies observed in the SM Yukawa interactions are common predictions in such setups.

A promising energy to search for short-distance 4F interactions at FCC-ee lies above the $Z$ peak, which is the main focus of this work. Beyond the $Z$-pole, the FCC-ee program includes three key energy stages [19]: $WW$ (163 GeV, 10 ab$^{-1}$), $Zh$ (240 GeV, 5 ab$^{-1}$), and $t\bar{t}$ (365 GeV, 1.5 ab$^{-1}$). These machine parameters provide an excellent opportunity to explore energy-enhanced 4F effects, with each stage offering competitive sensitivity due to the interplay between luminosity and energy, as we will demonstrate.

There are multiple $2 \rightarrow 2$ scattering processes to explore, including $e^+e^- \rightarrow b\bar{b}$, $c\bar{c}$, $s\bar{s}$, $jj$, $t\bar{t}$, $\tau^+\tau^-$, $\mu^+\mu^-$, and $e^+e^-$.[1] Many of these were previously studied at LEP-II [20, 21], but the significantly increased statistics at FCC-ee offer a unique opportunity to push precision measurements to new heights. To fully capitalize on this, precision observables such as the ratios of hadronic cross sections $R_a$ (and forward-backward asymmetries) are carefully chosen to ensure that SM predictions are precise enough. On the experimental side, flavor tagging is essential for achieving accurate measurements fully utilizing the expected luminosity. Recent advancements, particularly through the application of machine learning techniques [22], have significantly enhanced our ability to identify and separate different flavors with greater accuracy. Building on these developments, we design an optimized analysis strategy for measuring the $R_a$ ratios quantifying the projected sensitivity, as discussed in Section 2.

The interpretation of these observables in terms of NP begins by considering the tree-level effects of $2q2\ell$ and $4\ell$ operators. Additionally, a broader set of operators at the ultraviolet (UV) matching scale is accessible due to the SMEFT renormalization group (RG) running, driven by gauge couplings. This RG running enables the setting of competitive constraints on the third-family interactions and bosonic mediators, offering a direct comparison with complementary effects observed at the $Z$-pole. The bounds from the $Z$-pole run will also be examined in Section 3 for comparison.

In Section 4, we develop optimized search strategies for flavor-violating (FV) 4F interactions in $e^+e^- \rightarrow q_i\bar{q}_j$ and compare them with existing limits from meson decays, showing that decays are generically superior. For charged lepton FV, see [23]; for FV in Higgs and $Z$ decays, see [24].

Finally, as a practical application, in Section 5 we explore several models addressing the current flavor anomalies in $b \rightarrow s\ell^+\ell^-$ and $b \rightarrow c\tau\nu$ transitions. We demonstrate how FCC-ee can effectively probe the relevant parameter space in these models, providing valuable insight into these intriguing puzzles.

## 2 Fermion pair-production above the $Z$-pole

This section forms the core of our analysis. Here we focus on establishing precision observables above the $Z$-pole crucial for constraining flavor-conserving 4F interactions. For each of these, we develop optimized analysis strategies and derive the expected sensitivity achievable at FCC-ee.

---

[1]Here, we collectively denote with $j$ both $u$ and $d$ quarks, that with current tagging technology are difficult to disentangle.

## 2.1 Case study: $R_b$

We start with a simplified scenario where we aim to identify the bottom quark pairs in a sample of hadronic events. The counting experiment consists of three bins based on the number of $b$-tagged jets, with the mean number of events per bin given by

$$
\begin{aligned}
N(n_b = 2) &\equiv N_2 = N_{\text{tot}}[(\epsilon_b^b)^2 R_b + (\epsilon_j^b)^2 R_j], \\
N(n_b = 1) &\equiv N_1 = 2N_{\text{tot}}[\epsilon_b^b(1-\epsilon_b^b)R_b + \epsilon_j^b(1-\epsilon_j^b)R_j], \\
N(n_b = 0) &\equiv N_0 = N_{\text{tot}}[(1-\epsilon_b^b)^2 R_b + (1-\epsilon_j^b)^2 R_j].
\end{aligned}
\tag{1}
$$

Here,

$$
R_b = \frac{\sigma(e^+ e^- \to b\bar{b})}{\sum_{q=u,d,s,c,b} \sigma(e^+ e^- \to q\bar{q})},
\tag{2}
$$

is the ratio of cross sections for producing bottom-quark pairs relative to the total sum over all quark pairs. This ratio, as a function of the dijet invariant mass, can be accurately calculated within the SM [25], making it a clean observable for precision tests of the SM. We aim to estimate the relative precision $\delta_{R_b} \equiv \Delta R_b / R_b$ at FCC-ee operating *above* the $Z$ peak. The measurement will be performed at three collider energies corresponding to the $WW$, $Zh$, and $t\bar{t}$ thresholds.

The other ratio in Eq. (1) is $R_j = 1 - R_b$. In addition, $N_{\text{tot}} = \mathcal{L} \cdot \mathcal{A} \cdot \sigma(e^+ e^- \to q\bar{q})$ is the total number of dijet events *before* flavor tagging. Here, $\mathcal{L}$ is the expected luminosity, while $\mathcal{A}$ is the acceptance times efficiency of kinematical cuts selecting non-radiative events [26] with the dijet invariant mass close to the nominal collider energy. (Residual backgrounds, beyond $e^+ e^- \to jj$, will be discussed later.)

Flavor tagging plays a critical role in measuring the $R_b$ ratio precisely from the three bins in Eq. (1). The tagger efficiencies are denoted by $\epsilon_i^j$, indicating the probability of a flavor $i$ to be tagged as $j$. Important parameters here are $\epsilon_b^b$ (true positive) and $\epsilon_j^b$ (false positive). Tagger algorithms aim to minimize $\epsilon_j^b$ for a given $\epsilon_b^b$. This work utilizes the recently developed DeepJetTransformer from [22]. The tagger's ROC curves, showing $\epsilon_j^b$ as a function of $\epsilon_b^b$, are taken from Fig. 6 of the same reference. Finally, the negative rates used in Eq. (1) are by definition $\epsilon_b^j = 1 - \epsilon_b^b$ (false negative), and $\epsilon_j^j = 1 - \epsilon_j^b$ (true negative).

With three bins, we can float two additional variables besides $R_b$. This observation forms the basis of our strategy: determining the total number of events $N_{\text{tot}}$ and the true positive efficiency $\epsilon_b^b$ directly from the fit. As a result, the $R_b$ ratio, a theoretically clean observable, remains independent of the imprecise prior knowledge of $N_{\text{tot}}$ and $\epsilon_b^b$. Instead, the necessary input to the fit is the relative systematic uncertainty on $\epsilon_j^b$. A realistic estimate for FCC-ee is $\delta_\epsilon \approx 0.01$ as advocated in [24, 27].

The particle count in each bin follows a Poisson distribution with a mean from Eq. (1). However, the event numbers are expected to reach the millions (see later). Thus, the distributions can be approximated as Gaussians, simplifying the log-likelihood to

$$
-2\log L = \sum_i \frac{(N_i^{\text{exp}} - N_i)^2}{N_i^{\text{exp}}} + \frac{x^2}{(\delta_\epsilon)^2},
\tag{3}
$$

where the last term introduces a nuisance parameter $x$ for the systematic uncertainty on $\epsilon_j^b$, while simultaneously replacing $\epsilon_j^b \to \epsilon_j^b(1+x)$ in Eq. (1).

Using this likelihood, we can calculate the expected accuracy on $R_b$. Applying the Asimov approximation [28], the measured event count per bin is replaced by the expected count based

on nominal input values. The maximum likelihood estimators for $R_b$, $N_{\text{tot}}$, and $\epsilon_b^b$ are set to their nominal values, with uncertainties derived from the Hessian of $-2 \log L$. We find the variance to be

$$
\begin{aligned}
\left(\frac{\Delta R_b}{R_b}\right)^2 &= \frac{1 - \epsilon_b^b(2 - \epsilon_b^b(2 - R_b))}{N_{\text{tot}} R_b (\epsilon_b^b)^2} \\
&+ \frac{2(\epsilon_b^b - R_b(2 - \epsilon_b^b)(2\epsilon_b^b - 1))}{N_{\text{tot}} R_b^2 (\epsilon_b^b)^3} \epsilon_j^b \\
&+ \frac{4(R_b - 1)^2 (\epsilon_j^b)^2}{R_b^2 (\epsilon_b^b)^2} (\delta_\epsilon)^2 + \mathcal{O}\left((\epsilon_j^b)^2\right).
\end{aligned}
\tag{4}
$$

This equation is crucial for several reasons. The first term represents the statistical error from the true positive rate. The second term reflects the statistical error from the false positive rate, where reducing $\epsilon_j^b$ improves precision as expected. Both terms follow the typical scaling of $1/\sqrt{N}$ for counting experiments. The third term, a systematic error from uncertainty on the false positive rate, contributes a constant error to $R_b$ that does not decrease with more events, setting the ultimate precision limit.

The complete expression for $\Delta R_b / R_b$ features a non-trivial dependence on $\epsilon_b^b$, that can be minimized to find the optimal working point. To do so, we employ the ROC curves from [22] in a conservative manner, taking as $\epsilon_j^b$ the curve with worst mistag rate ($\epsilon_c^b$) and continuously extending it with a flat behavior in the region where the curve is not available. As a concrete benchmark, we focus on the $WW$ energy run, expected to deliver the most events. With a nominal integrated luminosity of $10 \, \text{ab}^{-1}$ [1] and the SM inputs $\sigma(\bar{e}e \to \text{hadrons}) \approx 34 \, \text{pb}$ and $R_b \approx 0.17$, we find that a precision of $2 \cdot 10^{-4}$ on $\Delta R_b / R_b$ is within the reach of FCC-ee. This is achieved for $\delta_\epsilon \simeq 0.01$, $\epsilon_j^b \simeq 10^{-3}$, and $\epsilon_b^b \simeq 0.65$, where the first and third terms in Eq. (4) contribute comparably, with the second term being subleading. In conclusion, we find that it is possible to almost reach the naïve statistical limit of $1/\sqrt{N_{\text{tot}} R_b}$. To fully leverage this finding for NP searches, the SM theory must match the experimental precision, which appears achievable.[2]

In addition to the primary backgrounds, one should consider the 4-quark background from $e^+ e^- \to VV$, particularly under kinematic configurations where the quark pairs are collimated. This background, identified by the OPAL collaboration [26] as the dominant one beyond $q\bar{q}$, introduces an additional contribution to $R_b$. To estimate the effect, in our statistical model we replace $R_b \to R_b'$ where $R_b' = R_b(1 + C)$. Therefore, the relative error on $R_b$ becomes $\delta_{R_b} = \sqrt{(\delta_{R_b'})^2 + (\Delta C)^2}$, where $\delta_{R_b'} \approx 10^{-4}$. To maintain precision, one requires $\Delta C = C \cdot \delta_C \lesssim 10^{-4}$. From Figure 1 in [26], $C \simeq 0.01$ is inferred, requiring $\delta_C \lesssim 0.01$ to achieve the necessary precision via Monte Carlo modeling.

## 2.2 $R_b$, $R_s$, and $R_c$ simultaneously

We now move to the simultaneous determination of the ratios $R_b$, $R_s$, and $R_c$. Here, we will use *orthogonal* bottom, strange, and charm taggers, which, for a given initial quark seed $q$, yield a single outcome: either a $b$-jet, $s$-jet, $c$-jet, or an untagged light jet ($j$). This divides dijet events into ten disjoint bins, each approximated as a Gaussian, with the mean number of events per bin given by

$$
N_{ij} = N_{\text{tot}} \sum_z \frac{2}{1 + \delta_{ij}} R_z \epsilon_z^i \epsilon_z^j,
\tag{5}
$$

---

[2]The present theory uncertainty on $R_b$ at the $Z$ peak is $10^{-4}$, see Table 10.5 in [29] and Table 6 in [30]. Interestingly, comparable experimental precision at the FCC-ee $Z$ peak can be achieved using the double-tag method, accounting for hemisphere correlations (see the talk). Systematic uncertainties clearly dominate this measurement.

where $i, j, z \in \{b, s, c, j\}$. The orthogonality of the efficiencies gives $\sum_z \epsilon_i^z = 1$, and similarly, $\sum_z R_z = 1$. The fit includes seven floating variables: $N_{\text{tot}}, R_b, R_s, R_c$, and the true positive efficiencies $\epsilon_b^b, \epsilon_s^s, \epsilon_c^c$, across 10 bins. As in the previous section, a 1% uncorrelated relative error $\delta_\epsilon$ is added to each false positive rate. The ROC curves are from Fig. 6 of [22], extended as described earlier, with false-positive rates conservatively set to $10^{-3}$ when they fall outside the plot range. The covariance matrix is derived from the Hessian of the log-likelihood using the Asimov approximation.

We find the best result at the $WW$ threshold, with optimal working points of $\epsilon_b^b = 0.67$, $\epsilon_s^s = 0.07$, and $\epsilon_c^c = 0.60$. This yields:

$$\frac{\Delta R_b}{R_b} = 1.7 \times 10^{-4}, \qquad \frac{\Delta R_s}{R_s} = 3.7 \times 10^{-3}, \qquad \frac{\Delta R_c}{R_c} = 1.4 \times 10^{-4},$$

$$\rho = \begin{pmatrix} 1 & -0.006 & -0.22 \\ -0.006 & 1 & -0.006 \\ -0.22 & -0.006 & 1 \end{pmatrix}, \tag{6}$$

where the last line is the correlation matrix. This remarkable result confirms the FCC-ee's ability to simultaneously determine hadronic ratios with unprecedented (sub)per-mille precision, improving upon the LEP-II determination by two orders of magnitude [31]. The resulting correlation between $R_b$, $R_s$, and $R_c$ is minimal. In Appendix A, we also report results at the $Zh$ and $t\bar{t}$ thresholds, which are slightly less precise due to the reduced total event numbers.

Among the hadronic ratios, $R_s$ has the least precise measurement due to frequent mistagging of light quarks as $s$-jets. To reduce this background, a lower $\epsilon_j^s$ rate is needed, calling for a working point with small $\epsilon_s^s$. The optimal balance between statistical and systematic errors occurs around $\epsilon_s^s \simeq 0.08$ and $\epsilon_j^s \simeq 0.004$. This working point, however, drastically reduced the signal event sample compared to the other two. Further improvements in $s$-tagging methodology, particularly in reducing $\epsilon_j^s$ and its uncertainty for large $\epsilon_s^s$, are needed to bring the precision of $\Delta R_s/R_s$ in line with the other measurements.

## 2.3 $R_t$

At $\sqrt{s} = 365$ GeV, plenty of top quark pairs will be produced, leading to distinct experimental signatures. Let us define the following ratio for convenience,

$$R_t = \frac{\sigma(e^+e^- \to t\bar{t})}{\sum_{q=u,d,s,c,b} \sigma(e^+e^- \to q\bar{q})}. \tag{7}$$

Setting $m_t = 172.5$ GeV yields a tree-level value of $R_t \simeq 0.11$. For 1.5 ab$^{-1}$ we find the relative statistical uncertainty $\Delta R_t/R_t \simeq 1.2 \times 10^{-3}$. The current experimental uncertainty on $m_t$, at the level of a few percent, prevents theoretical predictions from reaching the precision needed to match the statistical error. At FCC-ee, several runs are proposed between $\sqrt{s} = 340$ GeV and $\sqrt{s} = 345$ GeV to measure the top quark mass with a precision below 0.01% [1]. Using this as an input for $R_t$, we find that the statistical limit discussed earlier can essentially be reached by the time of the $\sqrt{s} = 365$ GeV run. For a detailed study of observables related to top quark production at future colliders, see [32–34]. In our study, $R_t$ is relevant only for a specific subset of operators involving $t_R$, as discussed in Section 3.

## 2.4 $R_\ell$

Another valuable set of observables is provided by the *leptonic* ratios

$$R_\ell = \frac{\sigma(e^+e^- \to \ell^+\ell^-)}{\sum_{q=u,d,s,c,b} \sigma(e^+e^- \to q\bar{q})}, \tag{8}$$

where $\ell = e, \mu, \tau$. Assuming the measurement is statistically limited, the projected relative uncertainty is given by

$$\frac{\Delta R_\ell}{R_\ell} = \sqrt{\frac{1}{N_\ell} + \frac{1}{N_{\text{tot}}}}. \tag{9}$$

Here, $N_\ell$ is the number of observed leptons pairs of flavor $\ell$, while $N_{\text{tot}}$ was defined in Section 2.1. This results in relative errors of $\Delta R_{\tau,\mu}/R_{\tau,\mu} = \{1.6, 3.5, 9.7\} \times 10^{-4}$ at the $WW$, $Zh$, and $t\bar{t}$ thresholds. Experimentally, leptons are easier to reconstruct and tag; however, the main challenge lies in achieving the required precision in the theoretical predictions since radiative corrections differ between the numerator and denominator.[3] Theoretical improvements are anticipated before the FCC-ee begins operation. We highlight a clean observable with negligible theoretical uncertainty: the ratio $R_{\tau/\mu} = \frac{\sigma(e^+e^- \to \tau^+\tau^-)}{\sigma(e^+e^- \to \mu^+\mu^-)}$. While an excellent test of lepton flavor universality (LFU), it becomes redundant when considered alongside $R_\ell$, since $R_{\tau/\mu} = R_\tau/R_\mu$. The LFU ratio becomes useful if theory errors on $R_\ell$ are underestimated.

The situation for $R_e$ is complicated by the forward singularity associated with the Bhabha scattering. To deal with this, we impose a kinematical cut $|\cos\theta| < 0.9$ following ALEPH and OPAL [20]. The higher event statistics result in relative experimental errors smaller than those for $R_{\tau,\mu}$. The theoretical uncertainty above the $Z$-pole for large-angle Bhabha scattering was around 0.5% at LEP [35, 36]. While theory uncertainty reductions for the $Z$-pole ratio at FCC-ee have been estimated [37], no equivalent studies exist for higher energies. Here, we assume a tenfold reduction (to 0.05%) at these energies, anticipating future work to address this issue. With this assumption, $R_e$ still remains limited by theory. Note that Bhabha scattering will be essential for precisely determining $\alpha_{\text{em}}$, reducing parametric uncertainty in various precision observables. The energy dependence of 4F effects enables the simultaneous determination of $\alpha_{\text{em}}$ and contact interactions by comparing different energy runs.

## 2.5 Production asymmetries

The forward-backward asymmetries, measured at LEP-II with few-percent precision [21], will also serve as important observables at FCC-ee. For final-state leptons and unpolarized electron-positron pairs, they are defined as

$$A_\ell = \frac{\sigma_F(e^+e^- \to \ell^+\ell^-) - \sigma_B(e^+e^- \to \ell^+\ell^-)}{\sigma_F(e^+e^- \to \ell^+\ell^-) + \sigma_B(e^+e^- \to \ell^+\ell^-)}, \tag{10}$$

where $\sigma_{F/B}$ denotes the integrated cross-section in the forward/backward hemisphere. For hadronic final states, they are defined in an analogous way.

Denoting by $N_F$ and $N_B$ the number of forward and backward events, related to the cross-section via $N_{F/B} = \mathcal{L} \cdot \mathcal{A} \cdot \sigma_{F/B}(e^+e^- \to \ell^+\ell^-)$, the statistical error associated to $A_\ell$ is given by

$$\left.\frac{\Delta A_\ell}{A_\ell}\right|_{\text{stat}} = \sqrt{\frac{4N_F N_B}{(N_F - N_B)^2(N_F + N_B)}}. \tag{11}$$

Employing the tree-level results for the forward/backward cross-sections given in Appendix B, we obtain a relative precision of $\Delta A_\ell/A_\ell = \{3.3, 8.8, 27\} \times 10^{-4}$ at the three reference energies. This nearly two-order-of-magnitude improvement over LEP-II assumes limitations are purely statistical. However, assessing the validity of this assumption requires dedicated experimental analysis. For instance, precise detector geometry is crucial for evaluating potential systematics

---

[3]For reference, Table 10.5 in [29] shows $\Delta R_\tau^Z/R_\tau^Z \simeq 5 \times 10^{-4}$ at the $Z$ pole.

in hemisphere association of final-state leptons. In the absence of such a study, we use the statistics-only uncertainty to explore how $A_\ell$ compares to $R_\ell$ in relevant cases for this paper.

In Appendix B, we compare $A_\ell$ and $R_\ell$ within the SMEFT framework, neglecting theory uncertainties, which are well below the statistical ones [20]. We find both observables yield similar new physics reach, with $R_\ell$ performing slightly better on the benchmark single operator scenarios. Given this and the lack of systematic uncertainty estimates for $A_\ell$, we focus on $R_\ell$ observables moving forward. However, $A_\ell$ remains essential to the future physics program at FCC-ee, as it provides complementary information, probing orthogonal directions in the SMEFT parameter space and lifting flat directions, as shown in Appendix B. Similar considerations apply to hadronic final states, warranting a dedicated study.

## 2.6 Competition: $Z$-pole, $W$-pole and $\tau$ decays

The $Z$-pole observables measured at LEP [21] have provided essential tests of the SM. The FCC-ee will enable even more precise tests, improving the relative precision of $Z$-pole observables by more than two orders of magnitude.

In our case, these observables serve as "competitors" in the sense that they offer alternative probes of flavor-conserving interactions. However, this term is intentionally misleading. In fact, the complementarity of these measurements is a key advantage of FCC-ee, enabling it to probe new physics in flavor-conserving interactions across multiple energy scales, as demonstrated later.

We collect in Tables 1 and 2 the current and projected uncertainties on the pole observables of interest for our work. The tables are taken from [4], in which the current determinations and prospects are sourced from [19, 21, 29, 30, 38]. In particular, Table 1 includes observables strictly associated to $Z$-pole physics: hadronic and leptonic branching ratios and decay asymmetries. We added prospects for the effective number of neutrino species $N_\nu$ from [30], where projections are provided for two different measurements. We selected the one with the strongest expected relative precision, improving the current precision by a factor of approximately 10.

Table 2 presents observables related to $W$-pole and $\tau$ decays, where $\tau$ particles are produced at the $Z$-pole. The $W$-pole observables include measurements of $W$-boson properties conducted at the $WW$ threshold, such as mass, total width, and leptonic branching ratios. The last two rows of Table 2 show projections for leptonic $\tau$ decays, including also decays into electrons taken from [30].

In Sections 3 and 5, we will use these "competitor" observables to compare their projected bounds with those derived from our $R_a, R_t, R_\ell$ observables[4] above the $Z$-pole, conveniently summarized in Table 3.

## 3 SMEFT interpretation: Four-fermion $\Delta F = 0$ operators

Adding higher dimensional operators to the SM

$$\mathcal{L}_{\text{SMEFT}} = \mathcal{L}_{\text{SM}} + \sum_{\mathcal{O}} C_{\mathcal{O}}\, \mathcal{O}\,, \tag{12}$$

allows the capture of generic short-distance effects.[5] The SMEFT provides a natural framework in which the results obtained in the previous section can be exploited to constrain BSM physics

---

[4]Here and in the following, we collectively denote with $R_a$ the observables $R_b, R_s, R_c$.

[5]Unless specified otherwise, the bounds reported in this paper refer to the scale $\Lambda_{\mathcal{O}}$ (in TeV), where $C_{\mathcal{O}} = \pm 1/\Lambda_{\mathcal{O}}^2$.

Table 1: Current relative errors and FCC-ee projections for the $Z$-pole observables used in this work. This table is adapted from [4], with data from [21, 29, 38]. To distinguish these observables from those above the $Z$-pole in Table 3, a superscript "$Z$" has been added.

| Observable | Curr. Rel. Err. $(10^{-3})$ | FCC-ee Rel. Err. $(10^{-3})$ |
|---|---|---|
| $\Gamma_Z$ | 2.3 | 0.1 |
| $\sigma_{\text{had}}^0$ | 37 | 5 |
| $R_b^Z$ | 3.06 | 0.3 |
| $R_c^Z$ | 17.4 | 1.5 |
| $A_{\text{FB}}^{0,b}$ | 15.5 | 1 |
| $A_{\text{FB}}^{0,c}$ | 47.5 | 3.08 |
| $A_b^Z$ | 21.4 | 3 |
| $A_c^Z$ | 40.4 | 8 |
| $R_e^Z$ | 2.41 | 0.3 |
| $R_\mu^Z$ | 1.59 | 0.05 |
| $R_\tau^Z$ | 2.17 | 0.1 |
| $A_{\text{FB}}^{0,e}$ | 154 | 5 |
| $A_{\text{FB}}^{0,\mu}$ | 80.1 | 3 |
| $A_{\text{FB}}^{0,\tau}$ | 104.8 | 5 |
| $A_e^Z$ | 14.3 | 0.11 |
| $A_\mu^Z$ | 102 | 0.15 |
| $A_\tau^Z$ | 102 | 0.3 |
| $N_\nu$ | 50 | 0.8 |

Table 2: Current error and FCC-ee projections for selected $W$-pole and $\tau$ observables employed in this work. Table taken from [4], with results from [19, 21, 29, 30, 38].

| Observable | Value | Error | FCC-ee Tot. |
|---|---|---|---|
| $\Gamma_W$ [MeV] | 2085 | 42 | 1.24 |
| $m_W$ [MeV] | 80350 | 15 | 0.39 |
| $\text{Br}(W \to e\nu)(\%)$ | 10.71 | 0.16 | 0.0032 |
| $\text{Br}(W \to \mu\nu)(\%)$ | 10.63 | 0.15 | 0.0032 |
| $\text{Br}(W \to \tau\nu)(\%)$ | 11.38 | 0.21 | 0.0046 |
| $\tau \to \mu\nu\nu(\%)$ | 17.39 | 0.04 | 0.003 |
| $\tau \to e\nu\nu(\%)$ | 17.82 | 0.04 | 0.003 |

in a model-independent manner. In this section, we put bounds on a set of 4F operators contributing to the observables in Section 2 at the tree-level or via SMEFT RG. These are listed in Table 4 and grouped according to three classes: semileptonic ($2q2\ell$), fully leptonic ($4\ell$) and four-quark operators ($4q$).

Table 3: Summary of FCC-ee projections for the relative precision of hadronic and leptonic ratios above the $Z$-pole derived in this work, with results for the three runs at $\sqrt{s} = 163, 240$ and $365\,\text{GeV}$ shown in separate columns. $R_e$ already includes our estimate for the theory error.

| Observable/FCC-ee Rel. Err. $(10^{-3})$ | $WW$ | $Zh$ | $t\bar{t}$ |
|:---:|:---:|:---:|:---:|
| $R_b$ | 0.17 | 0.36 | 0.96 |
| $R_s$ | 3.7 | 5.8 | 10 |
| $R_c$ | 0.14 | 0.27 | 0.69 |
| $R_t$ | - | - | 1.2 |
| $R_{\tau,\mu}$ | 0.16 | 0.35 | 0.97 |
| $R_e$ | 0.50 | 0.52 | 0.64 |

Our results are given in the Warsaw basis [8], at 95% confidence level, and are obtained by turning on one operator at a time. We include both tree-level and SMEFT RG effects, the latter being crucial in the absence of a direct tree-level contribution to our set of observables. In this case, the Wilson coefficient at the scale $\mu < \Lambda$ associated to an operator $\mathcal{O}_i$ reads, in the leading-log approximation,

$$C_i(\mu) = \sum_{i \neq j} \frac{\gamma_{ij}}{16\pi^2} \frac{\bar{c}}{\Lambda^2} \log\left(\frac{\mu}{\Lambda}\right), \tag{13}$$

where $\gamma_{ij}$ is the entry of the anomalous dimension matrix encapsulating the running of an operator $\mathcal{O}_j$ into $\mathcal{O}_i$, assuming $i \neq j$ and $C_j = \bar{c}/\Lambda^2$. We always take $\bar{c} = 1$, unless stated otherwise. The matrix $\gamma_{ij}$ has been computed at dimension six in [39–41] and conveniently implemented in DSixTools [42], which we employ here.

The two following subsections distinguish between operators in Table 4 contributing at tree-level or via SMEFT RG to our observables.

### 3.1 Tree-level

In this section, we focus on operators contributing at the tree-level to Table 3. These include the semileptonic and fully leptonic operators listed in Table 4. Where possible, we compare with existing bounds. For instance, crossing symmetry implies that high-$p_T$ Drell-Yan processes at the LHC provide complementary constraints on semileptonic operators.

**Semileptonic operators**

Semileptonic operators involving electrons are the most relevant for FCC-ee physics, as they contribute at tree-level to the $R_a, R_t, R_\ell$ observables introduced earlier. The details on how they contribute to these ratios are given in Appendix B; notably, the operators $\mathcal{O}_{\ell edq}, \mathcal{O}_{\ell equ}^{(1)}, \mathcal{O}_{\ell equ}^{(3)}$ are the only ones not interfering with the SM amplitude, thus featuring $\mathcal{O}(\Lambda^{-4})$ dependence in the observable and being more loosely constrained.

In Fig. 1 we report the projected FCC-ee bounds obtained in this work and compare them with the current bounds and the projected bounds from the HL-LHC. The LEP-II bounds are sourced from Table 8.13 from [21], while the Cesium bounds are calculated using the expressions and values given by [9,10,43]. The LHC bounds are taken from [44], while the HL-LHC

Table 4: The set of operators considered in this analysis is divided into three classes: semileptonic, fully leptonic, and four-quark operators.

| | |
|---|---|
| $\mathcal{O}_{\ell q}^{(1)}$ | $(\bar{\ell}_p \gamma_\mu \ell_r)(\bar{q}_s \gamma^\mu q_t)$ |
| $\mathcal{O}_{\ell q}^{(3)}$ | $(\bar{\ell}_p \gamma_\mu \tau^I \ell_r)(\bar{q}_s \gamma^\mu \tau_I q_t)$ |
| $\mathcal{O}_{eu}$ | $(\bar{e}_p \gamma_\mu e_r)(\bar{u}_s \gamma^\mu u_t)$ |
| $\mathcal{O}_{ed}$ | $(\bar{e}_p \gamma_\mu e_r)(\bar{d}_s \gamma^\mu d_t)$ |
| $\mathcal{O}_{\ell u}$ | $(\bar{\ell}_p \gamma_\mu \ell_r)(\bar{u}_s \gamma^\mu u_t)$ |
| $\mathcal{O}_{\ell d}$ | $(\bar{\ell}_p \gamma_\mu \ell_r)(\bar{d}_s \gamma^\mu d_t)$ |
| $\mathcal{O}_{qe}$ | $(\bar{e}_p \gamma_\mu e_r)(\bar{q}_s \gamma^\mu q_t)$ |
| $\mathcal{O}_{\ell equ}$ | $(\bar{\ell}_p^j e_r)(\bar{d}_s q_t^j)$ |
| $\mathcal{O}_{\ell equ}^{(1)}$ | $(\bar{\ell}_p^j e_r)\epsilon_{jk}(\bar{q}_s^k u_t)$ |
| $\mathcal{O}_{\ell equ}^{(3)}$ | $(\bar{\ell}_p^j \sigma_{\mu\nu} e_r)\epsilon_{jk}(\bar{q}_s^k \sigma^{\mu\nu} u_t)$ |
| $\mathcal{O}_{\ell\ell}$ | $(\bar{\ell}_p \gamma_\mu \ell_r)(\bar{\ell}_s \gamma^\mu \ell_t)$ |
| $\mathcal{O}_{\ell e}$ | $(\bar{\ell}_p \gamma_\mu \ell_r)(\bar{e}_s \gamma^\mu e_t)$ |
| $\mathcal{O}_{ee}$ | $(\bar{e}_p \gamma_\mu e_r)(\bar{e}_s \gamma^\mu e_t)$ |
| $\mathcal{O}_{qq}^{(1)}$ | $(\bar{q}_p \gamma_\mu q_r)(\bar{q}_s \gamma^\mu q_t)$ |
| $\mathcal{O}_{qq}^{(3)}$ | $(\bar{q}_p \tau^I \gamma_\mu q_r)(\bar{q}_s \tau_I \gamma^\mu q_t)$ |
| $\mathcal{O}_{qu}^{(1)}$ | $(\bar{q}_p \gamma_\mu q_r)(\bar{u}_s \gamma^\mu u_t)$ |
| $\mathcal{O}_{qd}^{(1)}$ | $(\bar{q}_p \gamma_\mu q_r)(\bar{d}_s \gamma^\mu d_t)$ |
| $\mathcal{O}_{uu}$ | $(\bar{u}_p \gamma_\mu u_r)(\bar{u}_s \gamma^\mu u_t)$ |
| $\mathcal{O}_{dd}$ | $(\bar{d}_p \gamma_\mu d_r)(\bar{d}_s \gamma^\mu d_t)$ |
| $\mathcal{O}_{ud}^{(1)}$ | $(\bar{u}_p \gamma_\mu u_r)(\bar{d}_s \gamma^\mu d_t)$ |

projections are calculated by centering the LHC bounds and scaling them,

$$\Lambda_{\text{HL-LHC}} = \sqrt[n]{\frac{\mathcal{L}_{\text{HL-LHC}}}{\mathcal{L}_{\text{LHC}}}} \Lambda_{\text{LHC}}. \tag{14}$$

Here, $\Lambda_{\text{LHC}}$ is the symmetrized LHC bound for a specific operator, computed by taking

$$((C_+ - C_-)/2)^{-1/2},$$

where $C_+$ and $C_-$ are the greatest and lowest values for the Wilson coefficient from [44]. For the LHC luminosity, we use $\mathcal{L}_{\text{LHC}} = 137\,\text{fb}^{-1}$ from the CMS experiment, as reported in [44]. For the HL-LHC luminosity, we use $\mathcal{L}_{\text{HL-LHC}} = 2 \times 3000\,\text{fb}^{-1}$.

The value of $n$ is either 4 or 8, depending on how an operator contributes to $pp \rightarrow e^+ e^-$ tails. For all operators having indices $11jj$ with $j = 2$ or 3, we use $n = 8$ due to the heavy quark parton density suppression, which makes the SM channel negligible. This is also true for a subset of the 1111 operators: $\Lambda_{\ell q}^{(1)}, \Lambda_{qe}, \Lambda_{\ell edq}, \Lambda_{\ell equ}^{(1)}, \Lambda_{\ell equ}^{(3)}$. For the first two operators $\Lambda_{\ell q}^{(1)}, \Lambda_{qe}$, this is due to the up and down parton densities conspiring in such a way that leads to cancellation at $\mathcal{O}(\Lambda^{-2})$, while others do not interfere with the SM, contributing only at $\mathcal{O}(\Lambda^{-4})$. The remaining 1111 operators interfere with the leading SM channels, so we use $n = 4$.

The FCC-ee bounds coming from $R_a, R_t, R_\ell$ above the $Z$-pole summarised in Table 3 are obtained by constructing the combined likelihood for the $WW, Zh$, and $t\bar{t}$ runs. Interestingly, the constraints feature an approximate constant behavior with the energy scale. For example, the bounds on $\Lambda_{qe,3311}$ read $\{17.8, 17.4, 16.5\}$ TeV considering the likelihood for the three energy runs separately. Combining them, one obtains the result reported in Fig. 1 and Appendix B. This peculiar fact is due to an interplay between the lower precision on $R_a, R_\ell$ as the center-of-mass energy increases and the increase with $s$ of the BSM contribution (see Appendix B). The product of the two factors conspire to deliver an approximately constant bound across the three reference energies. This implies that a possible new physics effect encoded in such operators will be consistently seen across multiple energy scales, leaving an unmistakable trace.

Bounds from the $Z$-pole, $W$-pole and $\tau$ decays arise due to the running of semileptonic operators into the well-known set of operators affecting the observables in Tables 1, 2, reported e.g. in Appendix C of [45]. In evaluating the RGEs, we keep only the leading contributions, namely the ones proportional to the SM gauge and top Yukawa couplings. The likelihood is built considering all the observables simultaneously. We find that $A_\ell^Z$, and in particular $A_e^Z$, provide the leading constraints. Indeed, any of the semileptonic operators considered in this section feature at least a U(1)$_Y$ running into $\mathcal{O}_{He,11}, \mathcal{O}_{H\ell,11}^{(1)}$, directly contributing to $A_e^Z$. In addition to these, the operator $\mathcal{O}_{\ell q}^{(3)}$ runs with SU(2)$_L$ bosons into $\mathcal{O}_{H\ell,11}^{(3)}$ and $\mathcal{O}_{\ell\ell,1221}$, which also alter the muon decay used to determine the Fermi constant and hence indirectly all the $Z$ boson decays into pairs of SM fermions (see also the discussion in the purely leptonic operator section). A bigger numerical factor in the RGEs, together with the fact that $g^2 > g'^2$, explains why the bounds on these operators are stronger.

The bar plots in Fig. 1 are split according to the quark flavor indices belonging to the first, second, or third generation, which we now discuss separately.

Operators involving first-generation quarks are currently constrained by atomic parity violation and from high-$p_T$ Drell-Yan data from LHC. The latter will benefit from the higher luminosity available at HL-LHC. For these operators, FCC-ee will provide competitive complementary constraints but is unlikely to yield significant improvements, as current and projected HL-LHC bounds are already quite strong.

Second-generation operators involving charm quarks are currently constrained by the $R_c$ measurement at LEP-II and from LHC high-$p_T$ Drell-Yan data, while the ones involving strange quarks are constrained at the LHC. The HL-LHC will just slightly improve the current bounds in an approximate uniform manner, as well as $Z$- and $W$-pole FCC-ee observables. Conversely, our FCC-ee observables in Table 3 will lead to a significant improvement, strengthening constraints by almost an order of magnitude.

Operators involving top and bottom quarks will benefit most from FCC-ee. Presently, these operators' scales are bounded to a few TeV only, but FCC-ee could improve this by an entire order of magnitude, probing scales up to 40 TeV. Operators involving right-handed down quarks are more loosely constrained by $Z$-pole physics due to the absence of additional $y_t^2$ running into Higgs-fermion current operators. In contrast, for operators involving quark doublet or up singlet, the $y_t^2$ running induces significant effects on the $Z$-pole observables. Notably, both the $Z$-pole and above-$Z$-pole runs will reach similar sensitivity on these operators. This type of new physics would thus leave a clear, consistent signal across multiple energy scales.

**Purely leptonic operators**

Four-lepton operators involving a pair of electrons contribute at tree-level to $R_\ell$ observables. The bounds we obtain are summarized in the bar plot of Fig. 2, in which we retain only operators interfering with the SM amplitude. Due to the symmetry properties of these operators,

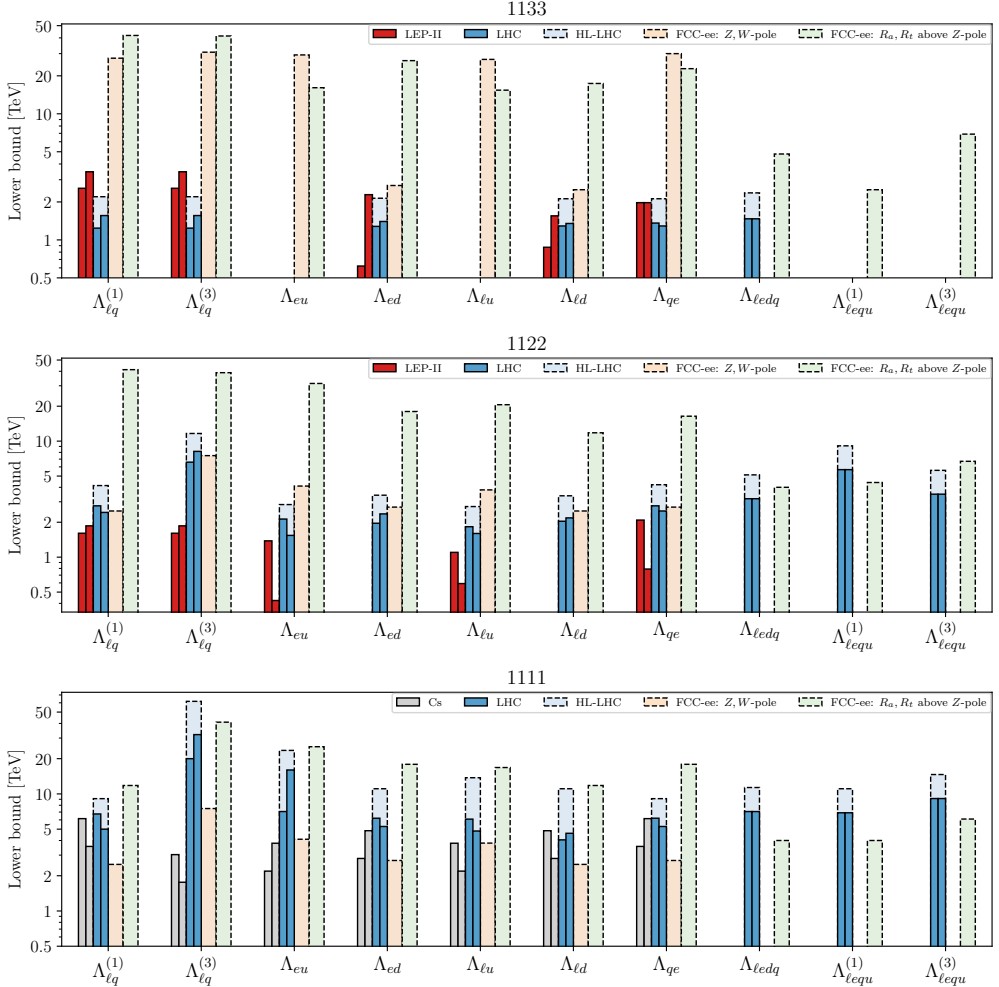

Figure 1: Current and projected constraints on the semileptonic operators in Table 4, considering one operator at a time. The three plots correspond to flavor-conserving operators involving first-, second-, and third-generation quarks. Only operators involving electrons are considered. When two bars are shown for the same observable, they indicate bounds for the negative and positive Wilson coefficients, respectively. The normalization in Eq. (12) is $C_{\mathcal{O}} = \pm 1/\Lambda_{\mathcal{O}}^2$. Note that the indices are inverted for $\mathcal{O}_{qe}$. For details, see Section 3.1.

there are far fewer possibilities for the flavor indices, and all the bounds fit nicely in just one plot. Along with our bounds, we also report the LEP-II bounds from [21] and the projected bounds from $Z, W$-pole physics and $\tau$ decays at FCC-ee.

Fig. 2 shows how the $R_\ell$ observables above the $Z$-pole at FCC-ee will lead to an almost homogeneous improvement compared to the current LEP-II bounds, probing up to ten times higher scales. This will even surpass the bounds from $Z, W$-pole observables and $\tau$-decays, to which this set of operators can contribute only at one-loop via a gauge running. Indeed, similarly to semileptonic operators, they run into Higgs-lepton current operators which contribute to $R_\ell^Z, A_{\mathrm{FB}}^{0,\ell}, A_\ell^Z$. The latter essentially dominate the bounds, aligning them around the same order of magnitude.[6] The only notable exceptions are associated with the operators $\mathcal{O}_{\ell\ell,1221}$ and $\mathcal{O}_{\ell\ell,1331}$, contributing at the tree-level to muon and tau decays. The first modifies the Fermi

---

[6]The operator $\mathcal{O}_{\ell\ell,1122}$ has slightly stronger bounds from $Z$-pole observables due to a $6g^2$ running into $\mathcal{O}_{\ell\ell,1221}$, which universally shifts the $Z$-boson couplings to fermions.

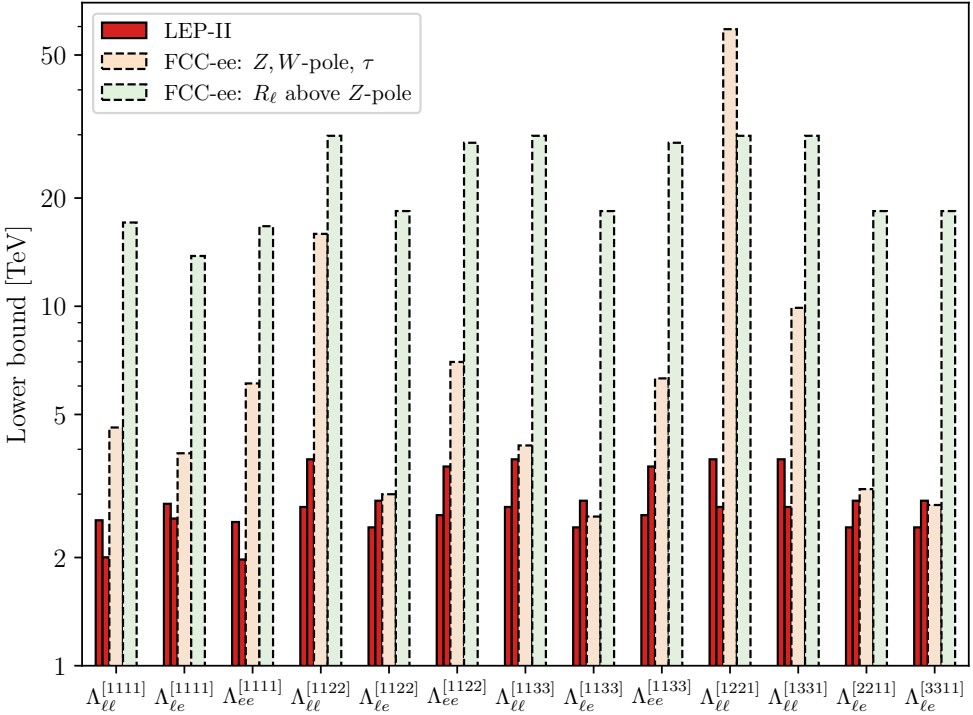

Figure 2: Current and projected constraints on the fully leptonic operators from Table 4, with one operator at a time, as in Fig. 1. For further details, see Section 3.1.

constant, whose effect propagates uniformly in all the observables, once again dominated by $A_\ell^Z$. Interestingly, the constraint from $m_W$ is somewhat weaker. The bound on $\mathcal{O}_{\ell\ell,1331}$ from $\tau$ decays is also strong, though unlike $\mathcal{O}_{\ell\ell,1221}$, its $Z$-pole bound is weaker than that from the above-$Z$-pole $R_\tau$. This is expected, as the projected precision improvement for the relevant observable is only about a factor of 13.

**Flavor universality and oblique corrections**

Flavor-universal new physics scenarios are motivated by many BSM models. Here, we consider two representative examples.

First, we turn on the vectorial operators in Table 4 one at a time, assuming flavor universality. The Wilson coefficients for different flavor indices are turned on simultaneously according to the U(3)$^5$-symmetric structure $C_{prst} = \delta_{pr}\delta_{st}/\Lambda^2$; for $\mathcal{O}_{\ell\ell}$, we distinguish the $\mathcal{O}_{\ell\ell}^D$ case, in which $C_{prst} = \delta_{pr}\delta_{st}/\Lambda^2$, and $\mathcal{O}_{\ell\ell}^E$ case, in which $C_{prst} = \delta_{pt}\delta_{rs}/\Lambda^2$, following [46]. The bounds we obtain are given in Table 5, in which we report both the results from the FCC-ee $Z, W$-pole observables with $\tau$ decays, and our FCC-ee above the $Z$-pole observables. Unsurprisingly, the bounds lie close to the naïve extrapolation of the strongest bound in Figs. 1 and 2 to the U(3)$^5$-symmetric structure. Similar reasoning also holds for the current LHC and projected HL-LHC bounds. In the semileptonic case, the constraints are essentially dominated by indices involving first-generation quarks, where high-$p_T$ Drell-Yan bounds apply. The current LHC bounds probe scales of $\mathcal{O}(5-15)$ TeV, with $\mathcal{O}_{\ell q}^{(3)}$ reaching even higher. HL-LHC will improve these to $\mathcal{O}(10-20)$ TeV. FCC-ee observables above the $Z$-pole will push the scales further to $\mathcal{O}(20-30)$ TeV, with the $O_{\ell q}^{(1),(3)}$ cases up to 50 TeV. Purely leptonic operators are currently constrained by LEP-II, with bounds of the order $\mathcal{O}(2-3)$ TeV. Here, the situation will improve dramatically at FCC-ee, pushing the scales up by a factor of 10. This improvement is

Table 5: Bounds on the operators in Table 4 in the flavor-universal $U(3)^5$ case. The bounds are reported at 95% CL. See Section 3.1 for details.

| $\Lambda$ [TeV] | $\Lambda_{\ell q}^{(1)}$ | $\Lambda_{\ell q}^{(3)}$ | $\Lambda_{eu}$ | $\Lambda_{ed}$ | $\Lambda_{\ell u}$ | $\Lambda_{\ell d}$ | $\Lambda_{qe}$ | $\Lambda_{\ell\ell}^{D}$ | $\Lambda_{\ell\ell}^{E}$ | $\Lambda_{\ell e}$ | $\Lambda_{ee}$ |
|---|---|---|---|---|---|---|---|---|---|---|---|
| FCC-ee $Z, W$-pole + $\tau$ | 33.0 | 43.1 | 32.9 | 6.0 | 30.7 | 5.7 | 34.6 | 12.8 | 59.4 | 8.5 | 13.2 |
| FCC-ee above $Z$-pole | 49.1 | 53.1 | 33.1 | 29.5 | 22.3 | 19.5 | 26.0 | 35.9 | 35.5 | 31.2 | 34.4 |

Table 6: Confidence intervals at 68% level on the oblique parameters $\hat{W}$ and $\hat{Y}$. See the text for details.

|  | $\hat{W} \times 10^5$ | $\hat{Y} \times 10^5$ |
|---|---|---|
| Current (LHC) | $[-19, 5]$ | $[-31, 14]$ |
| HL-LHC | $[-4.5, 6.9]$ | $[-6.4, 8.0]$ |
| FCC-ee pole observables | $[-3.1, 3.1]$ | $[-1.1, 1.1]$ |
| FCC-ee above the pole | $[-0.60, 0.60]$ | $[-2.2, 2.2]$ |

dominated by the ratios above the $Z$-pole, except in the case of $\mathcal{O}_{\ell\ell}^{E}$ due to the usual tree-level $G_F$ modification. Note that the limits presented in Table 5 are stronger than those obtained at $\pm 5$ GeV from the $Z$ peak [14]. While the vicinity of the $Z$ pole offers high statistics, the relative effect is much smaller, $\sigma_{\mathcal{O}}/\sigma_Z \sim C_{\mathcal{O}}(s - M_Z^2)$.

A second case of interest is given by the oblique corrections [47], specifically corrections to the gauge field propagators at $\mathcal{O}(p^4)$. These are captured by the $\hat{W}$ and $\hat{Y}$ parameters, entering the SMEFT lagrangian as [48]

$$\mathcal{L}_{\text{SMEFT}} \supset -\frac{\hat{W}}{4m_W^2}(D_\rho W_{\mu\nu}^a)^2 - \frac{\hat{Y}}{4m_W^2}(\partial_\rho B_{\mu\nu})^2 \,. \tag{15}$$

In the Warsaw basis, these operators are redundant and can be traded for flavor-universal combinations of the 4F operators in Table 4, as described, e.g., in [49]. In addition, they generate Higgs-fermion current operators and $O_{HD} = |H^\dagger D_\mu H|^2$ [5].[7] In Table 6, we report the marginalized confidence intervals at the $1\sigma$ level on the $\hat{W}, \hat{Y}$ parameters. The table includes the current bounds and the HL-LHC projected ones, taken from [49] (labeled SMEFT pdf), together with the FCC-ee projections obtained in this paper.[8] The former are given by high-$p_T$ Drell-Yan tails, which are impacted by semileptonic operators. FCC-ee pole observables are affected at the tree-level due to the presence of the Higgs-fermion current operators, the 4F operators $\mathcal{O}_{\ell\ell,1221}, \mathcal{O}_{\ell\ell,1331}, \mathcal{O}_{\ell\ell,2332}$ and $O_{HD}$. The constraints are dominated by $A_\ell^Z$ and $m_W$, and mark an improvement compared to HL-LHC. Interestingly, the resulting correlation is quite high, $\rho \simeq -0.91$, as all $A_\ell^Z$ feature the same parametric dependence on $\hat{W}$ and $\hat{Y}$, so that the flat direction is essentially lifted only by $m_W$. Finally, FCC-ee bounds employing the observables $R_a, R_t, R_\ell$ above the $Z$-pole will also provide a sizable improvement, due to the tree-level effect from semileptonic and purely leptonic operators. The most important contributions come from $R_{\tau,\mu}$ and $R_e$, and the correlation is also quite high, $\rho \simeq 0.59$. Again, this is due to the fact that the flat direction in $R_{\tau,\mu}$ is lifted only by $R_e$.

The FCC-ee projected bounds are presented in Fig. 3, with shaded areas indicating the allowed $1\sigma$ region. The directions probed by $Z$ and $W$-pole observables (blue region) and above-the-pole observables (red region) are nearly orthogonal. Consequently, their combined

---

[7]Other operators, such as $O_{\psi H}$, $O_{H\Box}$, and $O_H$, are also generated. These affect Higgs precision observables, some of which will be measured with high accuracy at FCC-ee [1]. However, they lead to much weaker constraints.

[8]In our projections, the $\hat{W}$ and $\hat{Y}$ parameters are evaluated approximately at the electroweak (EW) scale. For interpretation within a UV model, the SMEFT RG running to the UV scale should be included.

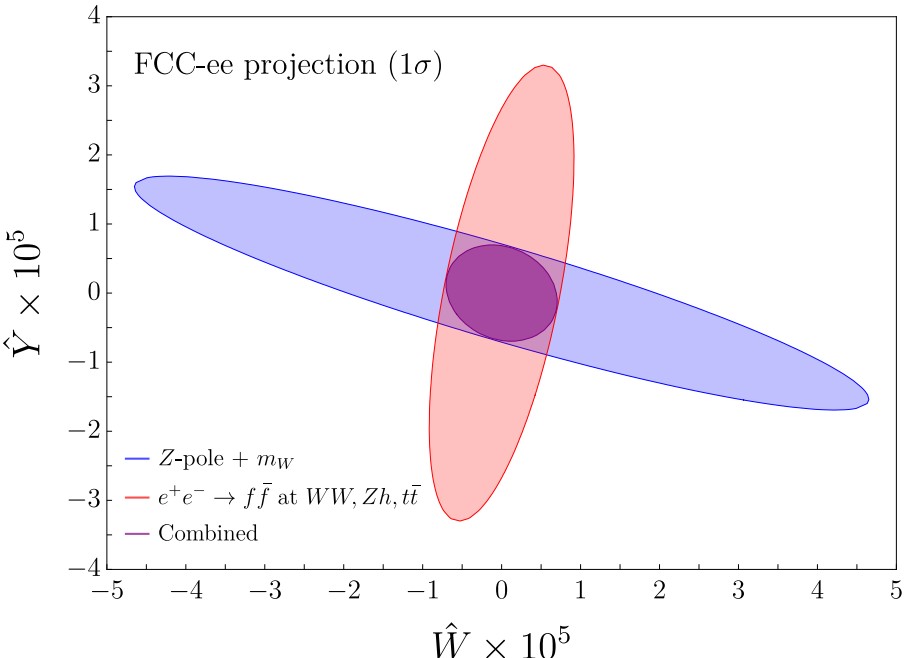

Figure 3: Projected FCC-ee $1\sigma$ contours on the oblique parameters $\hat{W}$ and $\hat{Y}$. The bounds from $Z, W$-pole observables and $R_a, R_\ell$ observables above the pole (denoted $e^+e^- \to f\bar{f}$) are first shown separately and then combined. See Section 3.1 for details.

fit (purple region) effectively disentangles the residual correlations within each individual fit, leading to impressive bounds. This also implies that the combined fit is largely insensitive to our estimated systematic uncertainties on $R_e$.

In summary, FCC-ee will improve the current bounds on the oblique parameters $\hat{W}$ and $\hat{Y}$ by at least an order of magnitude, with strong complementarity among electroweak precision observables at and above the pole.

## 3.2  RG effects

Four-quark operators and semileptonic, purely leptonic operators not involving electrons do not contribute to our ratio observables at the tree-level. Therefore, it is crucial to take into account the effects of renormalization. We focus on operators with flavor indices $prst = 3333$ (bounds reported in Table 7), since these are currently the least constrained operators (around a few TeV) and are also theoretically well-motivated, as discussed in the introduction. For comparison with the ratios above the $Z$-pole, we also include the expected bounds derived from the standard set of $Z$, $W$-pole, and $\tau$ observables.

Semileptonic operators can be constrained by $R_a, R_t, R_\ell$ ratios only via a gauge running, in which either a quark loop is closed and generates via a SM gauge interaction a pair of electrons (contributing then to $R_\tau$), or a $\tau$ loop is closed to generate a pair of electrons (directly contributing to $R_b, R_t$). Both contribute similarly to the global likelihood. Moving away from $prst = 3333$ flavor, operators involving muons instead of taus lead to identical bounds, as the projections on $R_\mu$ and $R_\tau$ are similar. The ones involving lighter quarks, instead, are expected to be comparably or slightly weaker constrained, giving direct contributions only to $R_{c,s}$.

For purely leptonic operators, the situation is even simpler. Closing a $\tau$ loop can only lead to other purely leptonic operators contributing to $R_\tau$. This analysis misses just the small set of operators simultaneously involving muons and taus. However, their running can con-

tribute only to $R_\tau$ and $R_\mu$. Given the similar expected precision on these two observables, we conclude that the bounds on these operators must be comparable to the ones of the purely third-generation case.

Finally, four-quark operators run into semileptonic ones by closing a quark loop and generating a pair of electrons via a SM gauge field. Hence, they contribute prominently to the $R_a, R_t$ observables. The constraints are essentially dominated by $R_b$, except for $\mathcal{O}_{uu}$ and $\mathcal{O}_{qu}$ for which $R_t$ becomes dominant. Moving away from the pure third-generation case, we expect the bounds to be weaker or, at most, comparable.

The results in Table 7 are all compatible with the expectation of a one-loop gauge running into the tree-level operators in Fig. 1, including the fact that the strongest bounds are on the operators involving an $SU(2)_L$ current due to $g^2 > g'^2$. The comparison with $Z, W$-pole and $\tau$ observables is also interesting. Purely leptonic operators run into $\mathcal{O}_{He,33}, \mathcal{O}_{H\ell,33}^{(1),(3)}$, which are then mostly constrained by observables involving $\tau$, in particular $A_\tau^Z$. Interestingly, $\mathcal{O}_{\ell\ell,3333}$ runs into $\mathcal{O}_{H\ell}^{(3)}$ and $\mathcal{O}_{\ell\ell,1331}$ with a $SU(2)_L$ coupling, meaning that for this operator also $\tau$ decays and $N_\nu$ play a relevant role, particularly the latter.[9] Among semileptonic operators, the strongest bounds are on those operators involving quark doublets and up singlets. These run into Higgs-lepton current operators with a top Yukawa instead of the usual gauge couplings. The most prominent observable is $A_\tau^Z$, as expected.

A similar analysis applies to four-quark operators. Curiously, however, the most important observable is $R_\mu^Z$, with the naïvely expected $R_b^Z$ as a close second. That is because the Higgs-quark current operators modify the $Z$ hadronic width, therefore indirectly affecting the precise $R_\ell^Z$ ratios. Finally, a notable exception to this discussion is $\mathcal{O}_{uu,3333}$, where the bound is due to a next-to-leading-log effect already noted in [4] (see also [50]). At one-loop, it runs into the operator $O_{Hu,33}$ with $y_t^2$, further enhanced by a big numerical coefficient. Integrating out the heavy top quark then results in an effective two-loop contribution that universally modifies the $Z$ coupling to all the SM fermions, see Fig. 12c of [45]. Then, the most important observables become $A_\ell^Z$, similarly to what happened with semileptonic operators in Section 3.1. Our result should be considered a rough estimate. A precise evaluation requires incorporating full next-to-leading-log effects by numerically solving the coupled RGE system, as done in [4], which provides a more reliable reference for this operator. Their result is compatible with ours.

# 4 Flavor violation

In contrast to the previous section, which concentrated on flavor-conserving but non-universal interactions, this section explores FCC-ee's potential to constrain flavor-violating 4F interactions. To complete earlier studies that investigated flavor violation in the tau [23] and top quark [51] sectors, we now turn our attention to the transitions $e^+e^- \to bs$, $e^+e^- \to bd$, and $e^+e^- \to cu$, which may exhibit an interesting interplay with low-energy flavor physics.

## 4.1 Search strategy

Our goal is to determine upper bounds on

$$R_{ij} = \frac{\sigma(e^+e^- \to q_i\bar{q}_j) + \sigma(e^+e^- \to q_j\bar{q}_i)}{\sum_{k,l=u,d,s,c,b} \sigma(e^+e^- \to q_k\bar{q}_l)} \, . \tag{16}$$

---

[9]The bound is, however, the weakest among purely leptonic operators due to a fortuitous cancellation in $A_\tau^Z$ among the induced Wilson coefficients.

Table 7: The 95% CL bounds at and above the $Z$-pole (at one-loop) on operators with flavor indices $prst = 3333$. See Section 3.1 for details.

| $\Lambda^{[3333]}$ [TeV] | FCC-ee $Z, W$-pole$+\tau$ | FCC-ee above $Z$-pole |
|---|---|---|
| $\Lambda^{(1)}_{\ell q}$ | 15.7 | 1.1 |
| $\Lambda^{(3)}_{\ell q}$ | 14.0 | 5.1 |
| $\Lambda_{eu}$ | 16.2 | 1.6 |
| $\Lambda_{ed}$ | 1.5 | 1.3 |
| $\Lambda_{\ell u}$ | 15.4 | 1.5 |
| $\Lambda_{\ell d}$ | 1.5 | 1.3 |
| $\Lambda_{qe}$ | 16.7 | 1.1 |
| $\Lambda_{\ell\ell}$ | 1.0 | 1.0 |
| $\Lambda_{\ell e}$ | 2.1 | 1.5 |
| $\Lambda_{ee}$ | 3.5 | 2.4 |
| $\Lambda^{(1)}_{qq}$ | 13.1 | 2.4 |
| $\Lambda^{(3)}_{qq}$ | 8.4 | 7.1 |
| $\Lambda^{(1)}_{qu}$ | 9.4 | 1.4 |
| $\Lambda^{(1)}_{qd}$ | 3.1 | 0.9 |
| $\Lambda_{uu}$ | 12.1 | 1.9 |
| $\Lambda_{dd}$ | 0.4 | 2.3 |
| $\Lambda^{(1)}_{ud}$ | 2.8 | 1.9 |

Taking into account the presence of flavor-violating ratios, the mean number of events in any bin is now given by a generalized version of Eq. (5):

$$N_{ij} = N_{\text{tot}} \sum_{k,l} \frac{1 + \delta_{kl}}{1 + \delta_{ij}} R_{kl} \epsilon^i_k \epsilon^j_l . \tag{17}$$

This expression implicitly assumes that the only sources of background are FCNC within the SM, which can be safely ignored, and the Drell-Yan processes studied in Section 2 in which one or two final state quarks get mistagged. Other sources of background are more difficult to model but may nevertheless play an important role.[10] Consistently with the aggressive approach taken in Section 2, we neglect them in the following. Later, we will show that even

---

[10]As in Section 2, an example is given by 4F process from pair production of $W$ bosons subsequently decaying in collimated jets. These can be misidentified as two quarks contributing to the $n_i = 1, n_j = 1$ bin if the angles between the jet axes are small enough. This background turns out to be quite important, although its analysis is non-trivial. To show this, let us focus on the $R_{uc}$ case. A possible contribution is due to the decay of $W \to ud$ and $W \to cs$, with collimated $us$ and $cd$ pairs. Assuming the former to be reconstructed as a light jet $j$ with $\epsilon^j_j$ probability, and the latter as $c$ with $\epsilon^c_c$ probability, the expected number of such background events is of the order $N_{\text{bckg},WW} \approx \epsilon^j_j \epsilon^c_c N_{WW} \text{Br}(W \to ud) \text{Br}(W \to cs) d\Omega$, where $d\Omega$ is a suppression factor encoding the probability of this angular coincidence happening. This factor should be estimated with a dedicated Monte Carlo simulation. As a brute approximation we can take the LEP-II estimation of the 4F background contribution to the number of $b\bar{b}$ events from [26], which result in $N_{WW} \text{Br}(W \to ud) \text{Br}(W \to cs) d\Omega \approx 10^{-2} N_{cc}$. This leads to $N_{\text{bckg},WW} \approx (10^{-3} - 10^{-2}) N_{cc}$. The background due to the mistag is instead $N_{\text{bckg, mistag}} \approx \epsilon^u_c \epsilon^c_c N_{cc} \approx 10^{-3} N_{cc}$. For the former to be subleading, it must be controlled with a relative precision better than $10^{-2}$, see Eq. (4).

Table 8: The bounds on $R_{ij}$ at 95% confidence level. See Section 4 for details.

| Energy | $ij$ | $R_{ij}$ |
|---|---|---|
| $WW$ | $bs$ | $2.80 \times 10^{-6}$ |
| | $bd$ | $3.44 \times 10^{-5}$ |
| | $cu$ | $5.28 \times 10^{-5}$ |
| $Zh$ | $bs$ | $6.37 \times 10^{-6}$ |
| | $bd$ | $6.58 \times 10^{-5}$ |
| | $cu$ | $1.10 \times 10^{-4}$ |
| $t\bar{t}$ | $bs$ | $1.79 \times 10^{-5}$ |
| | $bd$ | $1.53 \times 10^{-4}$ |
| | $cu$ | $2.70 \times 10^{-4}$ |

under this assumption, the new physics reach provided by these observables is inferior to that of low-energy probes of flavor violation.

We will verify a posteriori that the bounds on $R_{ij}$ we obtain is such that its contribution to the other bins is also very suppressed, necessarily involving again another mistag. This means that we can focus only on the the single bin of interest $n_i = 1, n_j = 1$. Then, the expected bound on $R_{ij}$ is simply obtained in the usual manner by computing the expected sensitivity in the Asimov approximation, $\mathrm{E}[S] = s/\sigma_b$, with expected number of events $s = N_{ij}$ and where $\sigma_b = (b + \sum_k \sigma_{b,k})^{1/2}$ denotes the sum in quadrature of the statistical uncertainty and the uncertainties associated to the parameters entering Eq. (17). These are fixed and taken from the previous fit. In this way, the bound on $R_{ij}$ has a simple expression:

$$R_{ij} < \frac{\sigma_b}{N_{\mathrm{tot}} \epsilon_i^i \epsilon_j^j} \cdot \Phi^{-1}(1 - \alpha), \tag{18}$$

where at 95% confidence level we have $\Phi^{-1}(0.95) = 1.645$. We stress that Eq. (18) is based on the Gaussian approximation, which can be easily verified to hold. Indeed, taking the $WW$ run as before, the number of background events for any bin is at least $2N_{\mathrm{tot}} R_z \epsilon_z^i \epsilon_z^j \approx \mathcal{O}(10^4)$.

Employing once again the ROC curves of [22], Eq. (18) can be minimized with respect to the tagging efficiencies to determine the optimal working point that provides the strongest bounds. Since the ROC curves of [22] cut off at $10^{-3}$, we choose to aggressively extend these curves further down to a minimum of $10^{-4}$. After digitizing the curve, we take the two points with the lowest mistag rates on the lin-log plot and perform a linear fit. Our analysis focuses on $R_{bs}$, $R_{bd}$, and $R_{cu}$, but not on $R_{sd}$, as disentangling $s$ and $d$ quarks is particularly challenging with the current tagging technologies. We minimize Eq. (18) for each $ij$ individually and report the results for all the three energy runs in Table 8. As expected, since the $b$-taggers have the smallest rate of mistags, we obtain the best bounds for $R_{bs}$, followed by $R_{bd}$ and $R_{cu}$. The minimization is non-trivial due to a competition between minimizing the mistags and maximizing the true positives. Both depend on the same free parameters $\epsilon_b^b, \epsilon_s^s, \epsilon_c^c$, but in opposite ways.

## 4.2 SMEFT interpretation

The semileptonic operators listed in Table 4 contribute at the tree-level to the flavor-violating processes $e^+ e^- \to q_i \bar{q}_j$. However, due to the strong suppression of the SM amplitude, interference effects are negligible, resulting in $\Lambda^{-4}$ scaling and thereby reduced sensitivity compared to flavor-conserving processes. Notably, all vector current operators contribute uniformly to

$R_{ij}$ ratios, given by

$$R_{ij} = \frac{s}{8\pi\sigma_{\text{had}}^{\text{SM}}} \sum_{\mathcal{O}} |\Lambda_{\mathcal{O},11ij}|^{-4}, \tag{19}$$

where $s$ represents the center-of-mass energy squared, and $\sigma_{\text{had}}^{\text{SM}}$ is the total hadronic cross-section $e^+e^- \to jj$ in the SM. We neglected tiny corrections to the $R_{ij}$ denominator.

Limits on the effective scale $|\Lambda_{\mathcal{O},11ij}|$ are set using the projected sensitivity on $R_{ij}$ ratios obtained above, combining data from runs above the $Z$ pole into a single likelihood. Due to the $\Lambda^{-4}$ dependence, stronger constraints are obtained at higher collider energies. While the precision on $R_{ij}$ is higher at lower energies due to increased statistics, it is not enough to compensate for the energy dependence.

The resulting limits we get are

$$|\Lambda_{1123}| > 16\,\text{TeV for } \mathcal{O}_{\ell q}^{(1)}, \mathcal{O}_{\ell q}^{(3)}, \mathcal{O}_{\ell d}, \mathcal{O}_{ed}, \mathcal{O}_{qe},$$

$$|\Lambda_{1113}| > 9.4\,\text{TeV for } \mathcal{O}_{\ell q}^{(1)}, \mathcal{O}_{\ell q}^{(3)}, \mathcal{O}_{\ell d}, \mathcal{O}_{ed}, \mathcal{O}_{qe}, \tag{20}$$

$$|\Lambda_{1112}| > 8.1\,\text{TeV for } \mathcal{O}_{\ell q}^{(1)}, \mathcal{O}_{\ell q}^{(3)}, \mathcal{O}_{\ell u}, \mathcal{O}_{eu}, \mathcal{O}_{qe},$$

at 95% confidence level.[11] If we choose not to extrapolate the ROC curves and instead fix the mistag rates to $10^{-3}$ outside the range of the plot in [22] as was done in Section 2, the $bs$ case worsens to 11 TeV while the other two bounds barely change.

Finally, we compare these bounds to the ones derived from meson decays via the crossing symmetry process $q_i \to q_j e^+e^-$. For $b \to s$ and $b \to d$ transition, a direct comparison with Table 4 in [52] shows the supremacy of meson decays. In addition, $c \to u$ are to be compared with [53]. Only operators involving right-handed up-type quarks provide competitive bounds, while all the others are subleading at FCC-ee.

In summary, even if a model predicts FV interactions, their contribution to the global fit for $R_{b,s,c}$ will be minimal due to the existing limits from hadronic decays (modulo cancellations). Therefore, FV effects can be safely neglected, allowing the focus to remain on the FC interactions.

## 5 Concrete model examples

This section demonstrates the impact FCC-ee would have on specific NP models. To be concrete, we analyze simplified models proposed to address the current anomalies in $B$-meson decays. While the NP origin of these anomalies remains uncertain, our analysis is instructive because it defines a clear target in the parameter space and highlights the interplay between high-$p_T$ colliders and indirect searches from $B$-meson decays.

The first two models focus on rare $b \to s\ell^+\ell^-$ transitions: a scalar leptoquark (Section 5.1) and a neutral vector $Z'$ (Section 5.2). The third model (Section 5.3) introduces a vector leptoquark to address simultaneously $b \to s\ell\ell$ and $b \to c\tau\nu$ anomalies. In all three cases, we find that the parameter space suggested by current anomalies will be probed at FCC-ee.

Flavor-changing neutral current transitions $b \to s\ell^+\ell^-$ where $\ell = e, \mu$ are sensitive indirect probes of heavy new physics reaching scales far beyond direct searches. The LHCb collaboration reported several anomalous measurements in $b \to s\mu^+\mu^-$ decays [54–58], most notably in the optimized angular observable $P_5'$. Interestingly, this has recently been independently confirmed by the CMS collaboration [59]. However, no deviation is observed in the lepton-flavor universality (LFU) ratios $R_{K^{(*)}}$ [60, 61] nor in $B_s \to \mu^+\mu^-$ decays [62, 63]. While the

---

[11]A similar analysis could be carried out also for $tc, tu$ at the $Zh, t\bar{t}$ runs. See [32–34].

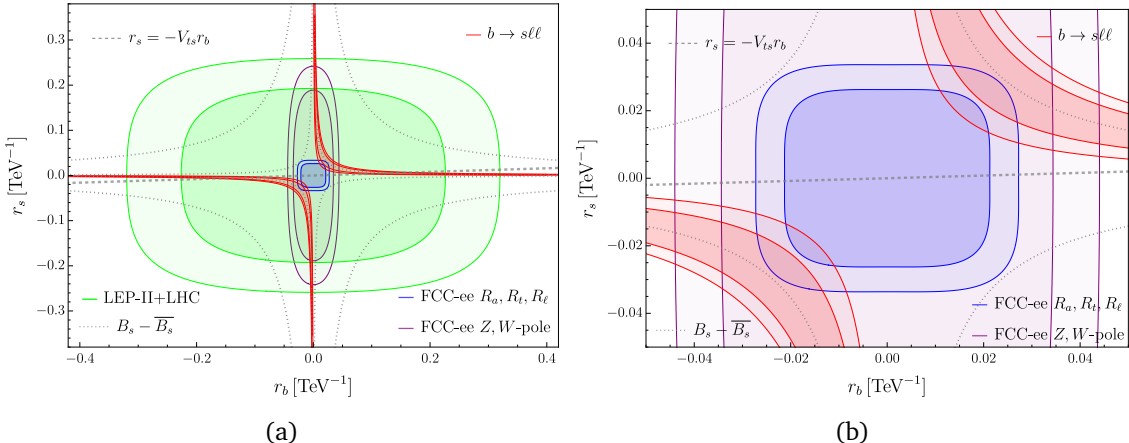

Figure 4: **Model I**: Scalar LQ for $b \to s\ell^+\ell^-$. The plot on the right is the zoomed-in version of the plot on the left. Darker and lighter shades correspond to the $1\sigma$ and $2\sigma$ confidence level intervals, respectively. The present fit to $b \to s\ell^+\ell^-$ decays prefers the red regions, while the green region is preferred by LEP-II $e^+e^- \to jj$ and LHC $pp \to \ell^+\ell^-$ high-$p_T$ tails. The dotted lines indicate the bounds from $B_s$ mixing for $M = 2$ TeV (outer) and $M = 40$ TeV (inner). The thick dashed line is $r_s = V_{cb}r_b$. Finally, the blue region shows the FCC-ee projections from hadronic $R_a$ ratios above the $Z$ peak, while the purple region shows the FCC-ee projections from the $Z$-pole observables and $W$ mass. For details, see Section 5.1.

deviations in the latter observables would be a clear sign of new physics given the reliable SM predictions, this is not the case for $P_5'$. Whether $P_5'$ is due to an unknown QCD effect or NP is a heated debate within the expert community, see e.g. [64–66]. Should the current experimental situation persist with more data from $B$-factories expected within this decade, it would be difficult to resolve this debate. Searching for correlated effects of NP at future colliders is therefore crucial, see [67].

The flavor-changing charged current transitions $b \to c\tau\nu$ occur at the tree-level in the SM. Therefore, the observed anomalies in $R_{D^{(*)}}$ ratios [68–73] suggest the need for significant contributions from NP. Nevertheless, consistent models can be constructed, as these transitions primarily involve third-generation interactions subject to weaker experimental constraints.

## 5.1 Model I: Scalar LQ for $b \to s\ell^+\ell^-$

The model proposed in [52] introduces lepton-flavor universal corrections to $b \to s\ell^+\ell^-$ transitions, mediated by leptoquarks at tree-level. To achieve this, a $U(2)_\ell$ flavor symmetry, acting on the light left-handed lepton doublets, is assumed to hold. Two scalar leptoquark fields, $S^\alpha \sim (\bar{\mathbf{3}}, \mathbf{3}, 1/3)$, where $\alpha = 1, 2$, form a doublet under $U(2)_\ell$. The relevant Lagrangian is:

$$\mathcal{L} \supset -M^2 S_\alpha^\dagger S^\alpha - \left( \lambda_i \, \bar{q}_i^c \ell_\alpha S^\alpha + \text{h.c.} \right), \tag{21}$$

where $\lambda_i$ (with $i = 1, 2, 3$) are input parameters of the model. As usual, the left-handed quark doublets $q_i$ are taken in the down-quark mass basis, $q^i = (V_{ji}^* u_L^j, d_L^i)^T$. The flavor components of the leptoquark doublet $S^\alpha = (S_e, S_\mu)^T$ are degenerate with mass $M$. In the following, we focus on $b \to s$ transitions, keeping only $\lambda_s \equiv \lambda_2$ and $\lambda_b \equiv \lambda_3$ nonzero and real.

Integrating out $S$ at the tree-level, one gets the SMEFT coefficients [52]

$$(C_{\ell q}^{(3)})_{\alpha\beta ij} = \frac{\delta_{\alpha\beta} \lambda_i^* \lambda_j}{4M^2}, \qquad (C_{\ell q}^{(1)})_{\alpha\beta ij} = \frac{3\delta_{\alpha\beta} \lambda_i^* \lambda_j}{4M^2}. \tag{22}$$

These can be directly matched to the operators $\mathcal{O}_9$ and $\mathcal{O}_{10}$ in the weak effective theory, correcting lepton-flavor universal coefficients

$$\Delta C_9^{\text{univ}} = -\Delta C_{10}^{\text{univ}} = \frac{r_s^* r_b}{2A_{\text{SM}}} = -0.46 \pm 0.11 \,, \tag{23}$$

where $r_i = \frac{\lambda_i}{M}$, while $A_{\text{SM}} = \frac{G_F \alpha_{\text{em}}}{\sqrt{2}\pi} V_{tb} V_{ts}^*$. The experimental value is derived from a combination of the global fit presented in [52] (Table 6) and the inclusion of the three low-$q^2$ $P_5'$ bins from the latest CMS analysis of $B^0 \to K^{*0}\mu^+\mu^-$ in [59] (Table 2), which slightly shifts the central value, indicating a larger tension. To remain conservative, we exclude the $[6-8.68]\,\text{GeV}^2$ bin.

In Fig. 4, we plot the region preferred by $b \to s\ell^+\ell^-$ data, as given in Eq. (23), with the darker and lighter red shadings representing the $1\sigma$ and $2\sigma$ confidence level intervals, respectively.

In addition, the same interactions contribute to $B_s - \overline{B}_s$ oscillations induced by one-loop box diagrams with a virtual exchange of $S^\alpha$. Following [74], the bound at 95% confidence level reads

$$|C_{B_s}^1| = \frac{5}{64\pi^2} \left| r_s^* r_b \right|^2 M^2 < 2 \times 10^{-5}\,\text{TeV}^{-2} \,. \tag{24}$$

The combination of Eq. (23) and Eq. (24) sets an upper limit on $M$. Benchmark values, $M = 2\,\text{TeV}$ and $40\,\text{TeV}$, are shown with thin-dotted outer and inner lines in Fig. 4, respectively.

This summarizes the key flavor constraints on the model. We now shift our focus to high-energy colliders and the bounds on flavor-conserving semileptonic interactions in Eq. (22) where $i = j$. The current constraints primarily arise from 1) high-$p_T$ $pp \to \ell^+\ell^-$ tails at the LHC and 2) $e^+e^- \to q\bar{q}$ scattering at LEP-II. For the first, we adopt the bounds presented in [44], while for the second, we interpret the analysis in [21]. Contact interactions provide an accurate approximation of the $t$-channel leptoquark exchange, even at the LHC, for $M \gtrsim$ current direct search limits. The combination is shown by the dark (light) green shading in Fig. 4 at $1\sigma$ ($2\sigma$). These constraints do not significantly challenge the model, only for extreme hierarchy in the parameters. For instance, the thick-dashed gray line representing $\lambda_s/\lambda_b = V_{cb}$, inspired by $U(2)_q$ flavor symmetries predicting the SM-like hierarchies, remains untested by these constraints.

Finally, we interpret our FCC-ee sensitivity analysis of $R_a, R_t, R_\ell$ from Sections 2.2, 2.3, 2.4 within the model's parameter space. The correction to these observables is predicted at the tree-level. Expressing the ratios as functions of $r_s$ and $r_b$, we construct the global likelihood by summing over the three energy runs using Eq. (6) and Appendix A. As shown in Section 4, the flavor-violating processes generated by the model can be safely neglected. The projected $1\sigma$ ($2\sigma$) bounds are shown in dark (light) blue. Figure 4b offers a zoomed-in view highlighting the FCC-ee reach. Remarkably, the FCC-ee measurement of the hadronic and leptonic ratios, in particular $R_b, R_s, R_c$, can probe this solution to the $b \to s\ell^+\ell^-$ anomalies and surpass the current bounds from $B_s$ mixing for weakly coupled new physics. This example demonstrates the significant leap the FCC-ee will make in probing flavor-conserving new physics, reaching a sensitivity level competitive with flavor-changing neutral current searches.

The RG running of Eq. (22) from the UV matching scale to the EW scale also introduces corrections to the $Z$ and $W$-pole observables, as discussed in Section 3. The resulting constraints are depicted in dark (light) pink at the $1\sigma$ ($2\sigma$) levels. As expected, the hadronic ratios above the $Z$ peak impose stronger limits. This is consistent with Fig. 1, including the fact that the constraint on $r_s$ is weaker compared to the one on $r_b$ due to the absence of the $y_t^2$ contribution in the RGE.

## 5.2 Model II: $Z'$ for $b \to s\ell^+\ell^-$

As an alternative to the leptoquark, we now consider a neutral vector mediator, $Z'_\mu \sim (\mathbf{1}, \mathbf{1}, 0)$. Lepton-flavor universal $Z'$ interactions, consistent with $R_{K^{(*)}}$, arise naturally in various UV completions; for a concrete example, see [52]. This offers an advantage over the leptoquark scenario, where introducing a doublet of states was necessary. For our purposes, we can work with an effective $Z'$ Lagrangian without concern for the details of the UV completion,

$$\mathcal{L} \supset g_{ij}\bar{q}_i\gamma_\mu q_j Z'^\mu + g_\ell(\bar{\ell}_\alpha\gamma_\mu\ell_\alpha + \bar{e}_\alpha\gamma_\mu e_\alpha)Z'^\mu. \tag{25}$$

For concreteness, we assume left-handed interactions in the quark sector and vector-like interactions for leptons, ensuring that $C_9$ is generated in the weak effective theory. Also, $g_{ij}$ is a hermitian flavor matrix while $g_\ell$ is real. We expect qualitatively similar conclusions for other models, such as [75,76].

Integrating out $Z'$ at the tree-level gives rise to several operators in the SMEFT. First, it generates $\mathcal{O}_{\ell q}^{(1)}$ and $\mathcal{O}_{eq}$, with the coefficients

$$(C_{\ell q}^{(1)})_{\alpha\beta ij} = (C_{eq})_{\alpha\beta ij} = -\delta_{\alpha\beta}\frac{g_\ell g_{ij}}{M^2}, \tag{26}$$

where $M$ is the $Z'$ mass. We assume $M \gtrsim v_{\text{EW}}$ such that the EFT provides a good description at FCC-ee. These can then be directly matched to the $O_9$ operator in the WET, which gives

$$\Delta C_9^{\text{univ}} = -\frac{r_\ell r_{sb}}{A_{\text{SM}}} = -0.88 \pm 0.17, \tag{27}$$

with $r_x = \frac{g_x}{M}$ and $A_{\text{SM}}$ defined below Eq. (23). $r_{sb}$ is considered real for simplicity. The experimental measurement is determined by the global $b \to s\ell^+\ell^-$ fit following the same procedure as in Section 5.1.

In addition, four-quark operators are generated, leading to $B_s$ meson mixing at the tree-level. The bound at 95% confidence level reads

$$|r_{sb}| < 0.0045\,\text{TeV}^{-1}\ (B_s\text{-mixing}). \tag{28}$$

Another significant constraint on the model comes from LEP-II measurements of $e^+e^- \to \ell^+\ell^-$. The operators generated at the tree-level are $\mathcal{O}_{\ell\ell}, \mathcal{O}_{\ell e}, \mathcal{O}_{ee}$, with the corresponding Wilson coefficients

$$C_{\ell\ell,11xx} = C_{ee,11xx} = C_{\ell e,iijj} = 2C_{\ell\ell,1111} = 2C_{ee,1111} = -r_\ell^2, \tag{29}$$

where $i, j = 1, 2, 3$ and $x = 2, 3$. We extract the current bounds from Table 8.13 in [26] incorporating both $e^+e^-$ and $\ell^+\ell^-$ data. This sets an upper limit at 95% confidence level

$$|r_\ell| < 0.20\,\text{TeV}^{-1} \quad \text{(LEP-II)}. \tag{30}$$

The three conditions in Eq. (27), Eq. (28) and Eq. (30) cover complementary regions in $(r_{bs}, r_\ell)$ plane. The combined fit to all three is depicted with dark (light) red shading for $1\sigma$ ($2\sigma$) in Fig. 5. This parameter space can consistently account for $b \to s\ell^+\ell^-$ anomalies while remaining compatible with the complementary constraints from $B_s$-meson mixing and LEP-II lepton pair production, making it a prime target for FCC-ee.

The first obvious improvement can be achieved by measuring the leptonic ratios $R_\ell$ at FCC-ee, which would substantially tighten the upper bound on $r_\ell$. The projected preferred region under the SM hypothesis is shown in dark (light) green for the $1\sigma$ ($2\sigma$) levels. Remarkably, these measurements alone will be enough to fully probe the entire parameter space of interest.

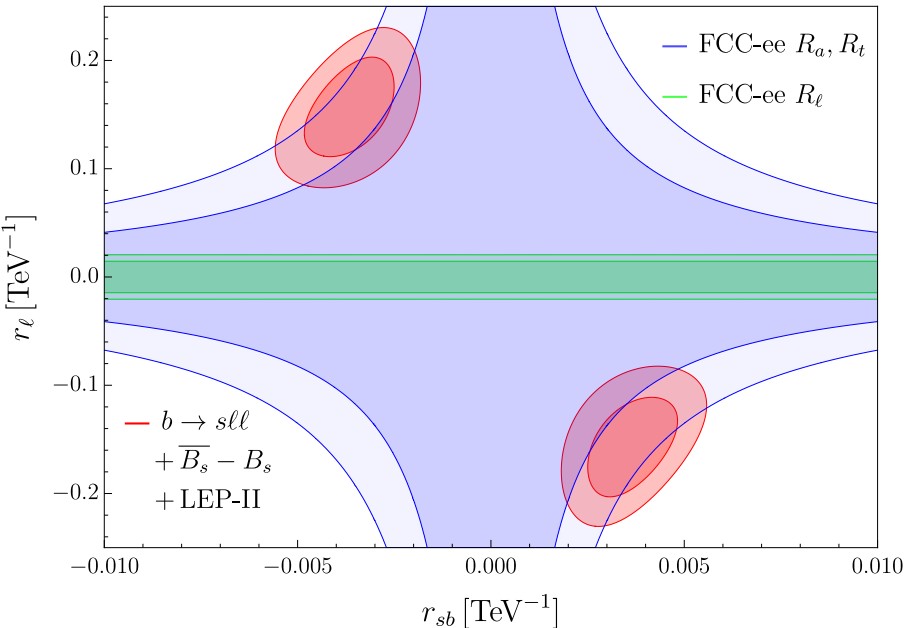

**Figure 5: Model II**: $Z'$ model for $b \to s\ell^+\ell^-$. The red contours represent the combined fit of $b \to s\ell^+\ell^-$, $B_s$-meson oscillations, and LEP-II measurements of leptonic ratios. The blue contours show FCC-ee projections for hadronic ratios, while the green contours correspond to FCC-ee projections for leptonic ratios. Darker and lighter contours indicate the $1\sigma$ and $2\sigma$ confidence levels, respectively. See Section 5.2 for details.

Finally, we close our discussion with the hadronic ratios $R_a$ at FCC-ee. While these do not provide a competitive direct bound on $r_{sb}$ as shown in Section 4, they set crucial bounds on $r_s$ and $r_b$. To relate those to $r_{sb}$, we employ the following inequality, which is fairly generic in UV completions [77],

$$(r_{sb}r_\ell)^2 \leq (r_s r_\ell)(r_b r_\ell) \leq \frac{1}{2}\left((r_s r_\ell)^2 + (r_b r_\ell)^2\right). \tag{31}$$

Using this inequality, the preferred regions from hadronic ratios in Fig. 5 are shown in dark and light blue for $1\sigma$ and $2\sigma$ confidence levels, respectively. Interestingly, these measurements will also (partially) probe the targeted parameter space indicated by $b \to s\ell^+\ell^-$ anomalies.

## 5.3 Model III: Vector LQ for $b \to c\tau\nu$ and $b \to s\ell^+\ell^-$

Consider a massive vector leptoquark field $U_\mu$ in the SM gauge representation $U_\mu \sim (\mathbf{3}, \mathbf{1}, 2/3)$. This single mediator model, with a TeV-scale mass and U(2)$^5$ flavor structure, is a well-regarded solution to both neutral and charged current $B$-anomalies [78–80]. The details of the UV completion [81–84] are not crucial for our discussion since we focus on the tree-level matching effects induced by the leptoquark itself. For simplicity, we consider a subset of models which predict $U_\mu$ interactions with the left-handed quark and lepton doublets only,[12]

$$\mathcal{L} \supset \frac{g_U}{\sqrt{2}}\beta_{i\alpha}\bar{q}_L^i\gamma^\mu l_L^\alpha U_\mu + \text{h.c.}, \tag{32}$$

with flavor indices $i$ for quarks and $\alpha$ for leptons. The quark and lepton doublets are defined in the down-quark and charged-lepton mass bases, respectively, such that $q_L^i = (V_{ji}^* u_L^j, d_L^i)^T$.

---

[12]Right-handed interactions lead to poorer compatibility between $R_{D^{(*)}}$ and $P_5'$.

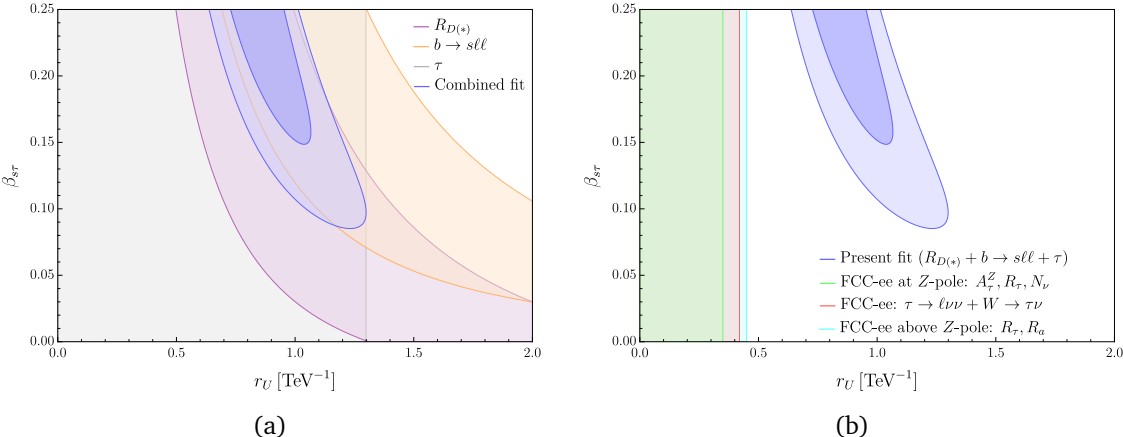

Figure 6: **Model III**: Vector LQ for $b \to c\tau\nu$ and $b \to s\ell^+\ell^-$. On the left panel, we break down the current low-energy constraints. In particular, the $2\sigma$ preferred regions are shown in purple for $R_{D^{(*)}}$, orange for the global $b \to s\ell^+\ell^-$ fit, and gray for LFU tests in $\tau$-decays. The dark blue (light blue) regions represent the combined fit at the $1\sigma$ ($2\sigma$) level. The right panel includes the same combined fit, along with FCC-ee constraints. For these, the $2\sigma$ preferred regions are shown in purple for the $Z$-pole observables $A^Z_\tau, R_\tau, N_\nu$, in gray for the $\tau \to \ell\nu\nu$ and $W \to \tau\nu$ decays, and orange for the above the $Z$-pole observables $R_\tau, R_a$. See Section 5.3 for details.

Here, $\beta_{i\alpha}$ is the flavor matrix, with $\beta_{b\tau} = 1$, real $\beta_{s\tau} = \mathcal{O}(V_{cb}) < 0.25$, and all other couplings being smaller. This assumption is consistent with a minimally-broken $U(2)^5$ flavor symmetry [16,17,85,86], which is motivated by the SM flavor puzzle and ensures that a UV completion remains consistent with current flavor constraints.

Integrating out the leptoquark at the tree-level, we get[13]

$$\mathcal{L}_{\text{SMEFT}} \supset -\frac{g_U^2}{4M_U^2} \beta_{i\alpha}\beta_{j\beta}^* \left[Q_{lq}^{(1)} + Q_{lq}^{(3)}\right]^{\beta\alpha ij}. \tag{33}$$

The two key parameters are $r_U = g_U/M_U$ and $\beta_{s\tau}$, and our main results are displayed in this parameter space in Fig. 6a. The dark blue (light blue) regions represent the combined fit to current data at the $1\sigma$ ($2\sigma$) confidence level, with the restriction $\beta_{s\tau} \leq 0.25$. This includes:

1. $b \to c\tau\nu$: The model predicts a shift in the strength of the left-handed weak interaction involving $\tau$, correcting the LFU ratios

$$R_{D^{(*)}} = R_{D^{(*)}}^{\text{SM}} \left|1 + \frac{r_U^2 v^2}{4}\left(1 + \beta_{s\tau}\frac{V_{cs}}{V_{cb}}\right)\right|^2, \tag{34}$$

where $v \approx 246\,\text{GeV}$ and $V_{ij}$ is the CKM matrix. In our fit, we input the latest HFLAV combination $R_D = 0.342 \pm 0.026$ and $R_{D^*} = 0.287 \pm 0.012$ with correlation $\rho = -0.39$ and the SM prediction $R_D^{\text{SM}} = 0.298$ and $R_{D^*}^{\text{SM}} = 0.254$. The $2\sigma$ range is shown with a purple shading in Fig. 6a.

2. $b \to s\ell^+\ell^-$: This model allows for both a lepton-universal contribution to $\Delta C_9^{\text{univ}}$ via an RG effect [90] and a tree-level lepton non-universal contribution, $\Delta C_9^\mu = -\Delta C_{10}^\mu$, arising from other leptoquark couplings $\beta_{s\mu}\beta_{b\mu}^*$, which are not the focus here.

---

[13]For details on the loop-level phenomenology in a gauged completion, see [87–89].

This latter contribution is constrained by $R_{K^{(*)}}$ and $B_s \to \mu^+\mu^-$ which agree with the SM. To account for this additional freedom, we rely on the global fit to all $b \to s\ell^+\ell^-$ data for a scenario that includes both contributions, as presented in Eq. (2.8) of [52]. As done previously in Sections 5.1 and 5.2, we combine this fit with the latest CMS analysis [59], and as a result, we obtain the lepton-universal coefficient $\Delta C_9^{\text{univ}} = -0.80 \pm 0.18$ after marginalizing over the other parameter. The model predicts

$$\Delta C_9^{\text{univ}} = \frac{\beta_{s\tau}}{V_{ts}^* V_{tb}} \frac{r_U^2 v^2}{6} \log\left(\frac{M_U^2}{m_b^2}\right), \tag{35}$$

where, for concreteness, we take $M_U = 2\,\text{TeV}$. The $2\sigma$ preferred region, highlighted in orange in Fig. 6a, shows remarkable consistency with $R_{D^{(*)}}$.

3. $\tau$ LFU: The RG mixing of Eq. (33) with $\beta\alpha ij = 3333$ into $\mathcal{O}_{H\ell}^{(3)}$ modifies the $W$-boson coupling to $\tau$, impacting LFU tests in $\tau$-decays [91]. Following [80], we find at $2\sigma$

$$r_U < 1.1\,\text{TeV}^{-1}. \tag{36}$$

This is represented by the gray shading in Fig. 6a.

The three constraints mentioned above are compatible and together define the parameter space of interest in blue in Fig. 6a, marking a clear target for future experiments. FCC-ee is especially well-positioned to explore this model through several complementary observables, shown in Fig. 6b. In the following, we will examine each observable contributing to this figure in detail. Since all of the effects are RG, we take $M_U = 2\,\text{TeV}$ for concreteness.

### $R_a$ and $R_\ell$ above the $Z$-pole

The RG mixing of Eq. (33) with $\beta\alpha ij = 3333$ into semileptonic and fully leptonic operators starting from the leptoquark mass scale down to the EW scale induces corrections to $R_a, R_t$, and $R_\ell$, respectively, as discussed in Section 3. The dominant effect arises from the gauge coupling $g$, generating sizable triplet operators $\mathcal{O}_{\ell q}^{(3)}$ and $\mathcal{O}_{\ell\ell}$. The overall impact is primarily driven by $R_\tau$, with subleading contributions from $R_b$. At the $2\sigma$ level, these set, respectively, the bounds $|r_U| < 0.47\,\text{TeV}^{-1}$ and $|r_U| < 0.78\,\text{TeV}^{-1}$. Combining them results in $|r_U| < 0.45\,\text{TeV}^{-1}$, corresponding to the preferred region shown in cyan in Fig. 6b.

### $Z$-pole observables

The RG mixing of Eq. (33) with $\beta\alpha ij = 3333$ into $[\mathcal{O}_{H\ell}^{(1),(3)}]^{33}$ leads to modifications in both $Z \to \tau\tau$ and $Z \to$ invisible ($N_\nu$). Notably, FCC-ee improvement in the $Z \to \tau\tau$ channel is significantly greater than in $N_\nu$. In the $\tau$ channel, however, contributions proportional to $y_t^2$ cancel, leaving only the $g^2$ term.

At the $2\sigma$ level, we obtain a combined bound of $|r_U| < 0.35\,\text{TeV}^{-1}$, corresponding to a scale $\Lambda_{\ell q, 3333}^{(1),(3)} \approx 5.7$ TeV. The preferred region is shown in green in Fig. 6b. This is about three times lower than the $\approx 15$ TeV bound from running the singlet or triplet third-generation operator (see Table 7). The $\chi^2$ analysis reveals that the bound is largely driven by $A_\tau^Z$, followed by $R_\tau^Z$ and $N_\nu$, with the latter being smaller by a factor of about 4.5. Interestingly, the contribution to $N_\nu$ is proportional to $y_t^2$, but the observable improves only by a factor of about 10. If $A_\tau^Z$ were removed, the projected bound would be relaxed to $|r_U| \lesssim 0.5$.

**$\tau$ and $W$ decays at FCC-ee**

The operator $\mathcal{O}^{(3)}_{\ell q,3333}$ runs into $\mathcal{O}^{(3)}_{H\ell}$, modifying $\tau$ and $W$ decays through a $y_t^2$-dependent contribution. The former effect is expressed as a correction to the branching fraction ratio $R_{\tau\alpha} = \text{Br}(\tau \to \alpha\nu\nu)/\text{Br}(\tau \to \alpha\nu\nu)_{\text{SM}}$, where $\alpha$ represents either muons or electrons. Writing $R_{\tau\alpha} = 1 + \delta R_{\tau\alpha}$, we obtain [92]

$$\delta R_{\tau\alpha} \simeq -\frac{m_t^2 N_c}{4\pi^2} \log \frac{m_U^2}{m_t^2} C^{(3)}_{\ell q,3333}, \tag{37}$$

with $C^{(3)}_{\ell q,3333} = -r_U^2/4$. Experimental data on $\tau \to \mu\nu\nu$ decays, with a branching ratio of 17.38% (and 17.82% for $\tau \to e\nu\nu$), enable precise comparisons with the SM prediction. FCC-ee is expected to improve the precision on $\text{Br}(\tau \to \mu\nu\nu, e\nu\nu)$ to $2 \times 10^{-4}$ [30], as reported in Table 2. Using the LFU ratio $(g_\tau/g_{\mu(e)})_\alpha = (R_{\tau\alpha}/R_{\mu e})^{1/2}$, the projected $2\sigma$ bound from FCC-ee is $|r_U| < 0.44\,\text{TeV}^{-1}$, an improvement by a factor of $\approx \sqrt{13}$ over the expected current bound of $|r_U| < 1.6\,\text{TeV}^{-1}$. However, since the current observed value is $(g_\tau/g_{\mu(e)})_\mu = 1.0012 \pm 0.0012$, the actual bound is around $r_U \simeq 1.1\,\text{TeV}^{-1}$.

The contribution to $\text{Br}(W \to \tau\nu)$ can be read off from Appendix C of [45]. Employing the expected bounds from FCC-ee, aiming at a relative precision on the branching ratio of $\approx 4 \times 10^{-4}$ [4] (see Table 2), leads at $2\sigma$ to $|r_U| \lesssim 0.64\,\text{TeV}^{-1}$. While subleading compared to the bound from $\tau$ decays, we stress that this would be already sufficient to probe the $U_\mu$ solution to the flavor anomalies at more than $2\sigma$. The preferred region from the combination of the two observables is shown in red in Fig. 6b, amounting to $|r_U| < 0.42\,\text{TeV}^{-1}$, and it covers the range of interest.

In conclusion, it is remarkable how many different electroweak observables will simultaneously probe the parameter space of the model. In addition, FCC-ee's flavor physics program ($B$ physics) will also provide further information, whose analysis is beyond the scope of this work.

# 6  Conclusions

The FCC-ee presents an exciting future for particle physics. In this work, we have highlighted its remarkable potential to probe fermion pair production above the $Z$-pole, demonstrating an unparalleled reach in exploring new physics that interacts with the SM in a flavor-conserving way.

In Section 2, we devised a strategy for measuring the hadronic ratios $R_b, R_c, R_s$ with exceptional precision, focusing on the $WW, Zh$, and $t\bar{t}$ runs above the $Z$ pole. By optimizing flavor tagging through recent machine learning advancements, we demonstrated FCC-ee's capability of improving LEP-II results by two orders of magnitude. In addition to hadronic ratios, we explored the top-quark ratio $R_t$, which benefits from FCC-ee's high-energy runs around the $t\bar{t}$ threshold, and the leptonic ratios $R_\ell$ for different lepton flavors. Forward-backward asymmetries offer orthogonal information to the cross-section ratios and are crucial for lifting flat directions in multi-dimensional fits and enhancing the overall sensitivity of the program.

In Section 3, we have interpreted our results in terms of 4F contact interactions in the SMEFT. A subset of semileptonic and fully leptonic operators are constrained at the tree-level by the hadronic and leptonic ratios, offering sensitivity to flavor-conserving new physics up to 50 TeV. With respect to current bounds, this corresponds to an improvement of more than an order of magnitude for second and third-generation quarks and all lepton generations. We also compared our bounds to those coming from FCC-ee observables at the $Z$- and $W$-poles, arising through RG effects within the SMEFT. Our findings suggest that the FCC-ee will offer multiple

complementary avenues to search for BSM physics. In Section 4, we derived bounds on flavor-violating 4-fermion interactions, showing a more modest improvement due to complementary constraints from meson decays.

Finally, we applied our bounds to three concrete models in Section 5 motivated by the present day $B$ anomalies in $b \to s\ell^+\ell^-$ and $b \to c\tau\nu$ transitions. Current complementary constraints still allow for explaining these anomalies in well-defined portions of the parameter space. We showed how FCC-ee would be able to cover these remaining regions entirely, either discovering or completely ruling out these scenarios.

The rich physics of fermion-pair production at FCC-ee, as explored in this study, offers an exciting glimpse into the future of particle physics, paving the way for unprecedented SM precision measurements and potential indirect discoveries of BSM. This work merely scratches the surface, inviting further theoretical refinements and dedicated experimental preparatory studies [93] as we anticipate the construction of this remarkable machine.

## Acknowledgments

We thank Lukas Allwicher, Javier Fuentes-Martín, Jakub Šalko, Aleks Smolkovič, Ben Stefanek and Felix Wilsch for useful discussion.

**Funding information** This work has received funding from the Swiss National Science Foundation (SNF) through the Eccellenza Professorial Fellowship "Flavor Physics at the High Energy Frontier," project number 186866.

## A $R_b$, $R_s$, and $R_c$ fit

In this Appendix, we report the result of Section 2.2 for other collider energies.

$Zh$: The optimal taggers' working point is $\epsilon_b^b = 0.71, \epsilon_s^s = 0.09$, and $\epsilon_c^c = 0.68$. The obtained standard deviations and correlations are:

$$\frac{\Delta R_b}{R_b} = 3.6 \times 10^{-4}, \qquad \frac{\Delta R_s}{R_s} = 5.8 \times 10^{-3}, \qquad \frac{\Delta R_c}{R_c} = 2.7 \times 10^{-4},$$
$$\rho = \begin{pmatrix} 1 & -0.008 & -0.25 \\ -0.008 & 1 & -0.023 \\ -0.25 & -0.023 & 1 \end{pmatrix}. \tag{A.1}$$

$t\bar{t}$: The optimal taggers' working point is $\epsilon_b^b = 0.78, \epsilon_s^s = 0.18$, and $\epsilon_c^c = 0.75$. The obtained standard deviations and correlations are:

$$\frac{\Delta R_b}{R_b} = 9.6 \times 10^{-4}, \qquad \frac{\Delta R_s}{R_s} = 1.0 \times 10^{-2}, \qquad \frac{\Delta R_c}{R_c} = 6.9 \times 10^{-4},$$
$$\rho = \begin{pmatrix} 1 & -0.012 & -0.27 \\ -0.012 & 1 & -0.034 \\ -0.27 & -0.034 & 1 \end{pmatrix}. \tag{A.2}$$

# B  SMEFT bounds

## B.1  Observables above the $Z$-pole

### $R_a, R_t$

In full generality, the contribution of the operators in Table 4 to the ratio

$$R_a = \sigma(\bar{e}e \to q_a \bar{q}_a)/\sigma(\bar{e}e \to \text{hadrons}),$$

can be written as

$$\frac{R_a}{R_a^{\text{SM}}} = \frac{1 + \sum_i c_{a,i} \Lambda_{a,i}^{-2} + \sum_{i,j} c_{a,ij} \Lambda_{a,i}^{-2} \Lambda_{a,j}^{-2}}{1 + \sum_{a',i} R_{a'}^{\text{SM}} c_{a',i} \Lambda_{a',i}^{-2} + \sum_{a',i,j} R_{a'}^{\text{SM}} \Lambda_{a',i}^{-2} \Lambda_{a',j}^{-2}}, \tag{B.1}$$

where $\Lambda_{a,i}^{-2}$ generically denotes the Wilson coefficient of an operator $i$ contributing to the ratio $R_a$, and $c_{a,i}, c_{a,ij}$ are the relative contributions to the cross-section normalized to the SM one. At order $1/\Lambda^2$ the previous expression can be rewritten as

$$\frac{R_a}{R_a^{\text{SM}}} - 1 = (1 - R_a^{\text{SM}}) \sum_i c_{a,i} \Lambda_{a,i}^{-2} - \sum_{a' \neq a, i} R_{a'}^{\text{SM}} c_{a',i} \Lambda_{a',i}^{-2}, \tag{B.2}$$

which illustrates how operators not directly involving $q_a$ nevertheless contribute to $R_a$ (although suppressed by $R_{a'}$). The coefficients $c_{a,i}$ can be easily determined at tree-level by computing the associated cross-section. For these, we find (cross-checking with the results in Appendix A of [18])

$$c_{a,i} = \pm \frac{2(g_{\text{SM},m_1,m_2}^{ae}(s))^2}{\sum_{m_1,m_2} (g_{\text{SM},m_1,m_2}^{ae}(s))^4} s, \tag{B.3}$$

where $-$ is only for $c_{u,\bar{i}}$ with $\bar{i} = O_{\ell q}^{(3)}$ and $+$ is for all the other vectorial semileptonic operators in Table 4. With $u(d)$ we denote here all up-type (down-type) quarks. The couplings in this expression are defined as

$$(g_{\text{SM},m_1,m_2}^{ae}(s))^2 \equiv e^2 Q_a Q_e + \frac{g_Z^{a_{m_1}} g_Z^{e_{m_2}}}{1 - m_Z^2/s}, \tag{B.4}$$

with $m_i = L, R$ and $a = u, d$. $Q_a, Q_e$ are the electric charges of $q_a$ and of the electron, and $g_Z^f$ are the couplings of the SM fermions to the $Z$ boson as given in [18]. In terms of $(g_{\text{SM},m_1,m_2}^{ae}(s))^2$, $R_a^{\text{SM}}$ can be easily expressed as

$$R_a^{\text{SM}} = \frac{\sum_{m_1,m_2} (g_{\text{SM},m_1,m_2}^{ae}(s))^4}{\sum_{a',m_1,m_2} (g_{\text{SM},m_1,m_2}^{a'e}(s))^4}. \tag{B.5}$$

The last three semileptonic operators in Table 4 do not interefere with the SM. Nevertheless, an expression analogous to Eq. (B.2) holds with the replacement $\Lambda_{a,i}^2 \to \Lambda_{a,i}^4$ and coefficients

$$c_{a,i} = \frac{s^2}{\sum_{m_1,m_2} (g_{\text{SM},m_1,m_2}^{ae}(s))^4} \times \left(\frac{3}{4}, \frac{3}{4}, 4\right), \tag{B.6}$$

for $(\mathcal{O}_{\ell edq}, \mathcal{O}_{\ell equ}^{(1)}, \mathcal{O}_{\ell equ}^{(3)})$.

The case of $R_t$ is exceptional, as we cannot neglect corrections due to the finite mass of the top quark. The analytical expression at tree-level retaining $m_t$ is lengthy; for our scopes, it is sufficient to report the values of $c_{t,i}$ at $\sqrt{s} = 365$ GeV:

$$c_{t,i} = (-1.25, 1.25, -0.86, 0, -0.79, 0, -0.47) \text{ TeV}^2\,, \tag{B.7}$$

for $\left(\mathcal{O}_{\ell q}^{(1)}, \mathcal{O}_{\ell q}^{(3)}, \mathcal{O}_{eu}, \mathcal{O}_{ed}, \mathcal{O}_{lu}, \mathcal{O}_{ld}, \mathcal{O}_{qe}\right)$. In the calculation we have set $m_t = 172.5$ GeV, and here and everywhere else we used as input $\alpha_{\text{EM}} = 1/128, m_Z = 91.19$ GeV and $s_W^2 = 0.231$.

**$R_\ell$**

Contributions to $R_\ell$ can be described in a similar manner. The main differences compared to Eq. (B.2) are $i$) the sum in the first term on the right-hand side runs only over leptonic operators, and the $(1-R_a^{\text{SM}})$ factor should be removed $ii$) the sum in the second term runs only over semileptonic operators, with $R_{a'}$ being the hadronic ratios only. Concerning the operators $(\mathcal{O}_{\ell\ell,11xx}, \mathcal{O}_{\ell\ell,1xx1}, \mathcal{O}_{\ell e,11xx}, \mathcal{O}_{\ell exx11}, \mathcal{O}_{ee,11xx})$, with $x = \mu, \tau$, the associated coefficients read

$$c_{\ell,i} = \pm \frac{2(g_{\text{SM},m_1,m_2}^{ee}(s))^2}{\sum_{m_1,m_2}(g_{\text{SM},m_1,m_2}^{ee}(s))^4} s\,, \tag{B.8}$$

where the minus sign is there just for $\mathcal{O}_{\ell\ell,1xx1}$. The operator $\mathcal{O}_{\ell e,1xx1}$ does not interfere with SM and its $\Lambda^{-4}$ coefficient reads $c_{\ell,i} = 3s^2/\sum_{m_1,m_2}(g_{\text{SM},m_1,m_2}^{ee}(s))^4$. As for the Bhabha scattering, we numerically computed the coefficients of $(\mathcal{O}_{\ell\ell,1111}, \mathcal{O}_{\ell e,1111}, \mathcal{O}_{ee,1111})$ with the cut described in the text. We obtain $c_{e,i} = (-0.05, -0.05, -0.05), (-0.13, -0.09, -0.13), (-0.32, -0.20, -0.31)$ TeV$^2$ for the three reference energies. We stress again that also semileptonic operators contribute to $R_\ell$ by modifying the hadronic cross-section, with associated coefficients $c_{a,i}$ given by (B.3).

**$A_\ell$**

Parametrizing the differential cross-section for the $e^+e^- \to \ell^+\ell^-$ scattering with $\ell = \mu, \tau$ as [18]

$$\frac{d\sigma}{dt} = \frac{1}{16\pi s^4}(t^2 X + u^2 Y)\,, \tag{B.9}$$

then

$$\sigma_F(e^+e^- \to \ell^+\ell^-) = \frac{1}{384\pi s}(X + 7Y)\,,$$
$$\sigma_B(e^+e^- \to \ell^+\ell^-) = \frac{1}{384\pi s}(7X + Y)\,, \tag{B.10}$$
$$A_\ell = \frac{3}{4}\frac{Y - X}{Y + X}\,.$$

In the SM, at tree-level $X = ((g_{\text{SM},LR}^{ee}(s))^4 + g_{\text{SM},RL}^{ee}(s))^4)$ and $Y = ((g_{\text{SM},LL}^{ee}(s))^4 + (g_{\text{SM},RR}^{ee}(s))^4)$. SMEFT corrections to this expression can be encoded, at $\mathcal{O}(\Lambda^{-2})$, as

$$X = X^{\text{SM}}\left(1 + \sum_i c_i^X \Lambda_i^{-2}\right)\,,$$
$$Y = Y^{\text{SM}}\left(1 + \sum_i c_i^Y \Lambda_i^{-2}\right)\,. \tag{B.11}$$

For the operators $(\mathcal{O}_{\ell\ell,11xx}, \mathcal{O}_{\ell\ell,1xx1}, \mathcal{O}_{\ell e,11xx}, \mathcal{O}_{\ell e,11xx}, \mathcal{O}_{ee,11xx})$ we get

$$
\begin{aligned}
c_a^X &= \frac{s}{2(g_{\text{SM},LR}^{ee}(s))^4}\left(0, 0, 2(g_{\text{SM},LR}^{ee}(s))^2, 2(g_{\text{SM},LR}^{ee}(s))^2 0\right), \\
c_a^Y &= \frac{s}{(g_{\text{SM},LL}^{ee}(s))^4 + (g_{\text{SM},RR}^{ee}(s))^4}\left(2(g_{\text{SM},LL}^{ee}(s))^2, -2(g_{\text{SM},LL}^{ee}(s))^2, 0, 0, 2(g_{\text{SM},RR}^{ee}(s))^2\right),
\end{aligned}
\tag{B.12}
$$

where we used the fact that $(g_{\text{SM},LR}^{ee}(s))^2 = (g_{\text{SM},RL}^{ee}(s))^2$.

## B.2 Numerical results

The combined bounds (meaning including the $WW, Zh$ and $t\bar{t}$ runs) at 95% confidence level plotted in Figs. 1, 2 are reported in Tables 10, 11. We include both bounds from the $R_a, R_t, R_\ell$ observables above the $Z$-pole and from the $Z, W$-pole observables mentioned in Section 2.6.

In Table 9 we compare the 95% confidence level bounds from $A_\ell$ to those from $R_\ell$ for a set of purely leptonic operators involving muons and taus. The bounds are similar, although the ones from $A_\ell$ are systematically weaker in this single-operator benchmark case.

Table 9: Combined bounds on purely leptonic operators at tree-level from $A_\ell$ and $R_\ell$. Here $x = 22, 33$.

| Observable/$\Lambda$ [TeV] | $\Lambda_{\ell\ell,11xx}(\Lambda_{\ell\ell,1xx1})$ | $\Lambda_{\ell e,11xx}(\Lambda_{\ell e,xx11})$ | $\Lambda_{ee,11xx}$ |
|:---:|:---:|:---:|:---:|
| $R_\ell$ | 29.8 | 18.4 | 28.5 |
| $A_\ell$ | 11.7 | 18.1 | 11.2 |

Table 10: Comparison between the combined bounds from $R_a, R_t, R_\ell$ above the $Z$-pole as well as from the $Z$-pole, $W$-pole and $\tau$ decay observables. The bounds are at 95% CL and are the same entering Fig. 1.

| prst/ $\Lambda$ [TeV] | | $\Lambda_{\ell q}^{(1)}$ | $\Lambda_{\ell q}^{(3)}$ | $\Lambda_{eu}$ | $\Lambda_{ed}$ | $\Lambda_{\ell u}$ | $\Lambda_{\ell d}$ | $\Lambda_{qe}$ |
|:---:|:---|:---:|:---:|:---:|:---:|:---:|:---:|:---:|
| 1133 | Above $Z$-pole | 41.8 | 41.4 | 16.1 | 26.4 | 15.4 | 17.4 | 22.8 |
| | $Z, W$-pole $+ \tau$ | 27.6 | 30.8 | 29.3 | 2.7 | 27.0 | 2.5 | 30.0 |
| 1122 | Above $Z$-pole | 41.3 | 38.9 | 31.4 | 18.0 | 20.6 | 11.8 | 16.4 |
| | $Z, W$-pole $+ \tau$ | 2.5 | 7.5 | 4.1 | 2.7 | 3.8 | 2.5 | 2.7 |
| 1111 | Above $Z$-pole | 11.8 | 41.0 | 25.3 | 17.9 | 16.8 | 11.8 | 17.9 |
| | $Z, W$-pole $+ \tau$ | 2.5 | 7.5 | 4.1 | 2.7 | 3.8 | 2.5 | 2.7 |

Table 11: Comparison between the combined bounds from $R_\ell$ above the $Z$-pole as well as from the $Z$-pole, $W$-pole and $\tau$ decay observables. The bounds are at 95% CL and are the same entering Fig. 2. The entries left blank correspond to redundant flavor indices.

| prst/$\Lambda$ [TeV] | | $\Lambda_{\ell\ell}$ | $\Lambda_{\ell e}$ | $\Lambda_{ee}$ |
|---|---|---|---|---|
| 1111 | Above $Z$-pole | 17.1 | 13.8 | 16.7 |
| | $Z, W$-pole + $\tau$ | 4.6 | 3.9 | 6.1 |
| 1122 | Above $Z$-pole | 29.8 | 18.4 | 28.5 |
| | $Z, W$-pole + $\tau$ | 15.9 | 3.0 | 7.0 |
| 1133 | Above $Z$-pole | 29.8 | 18.4 | 28.5 |
| | $Z, W$-pole + $\tau$ | 4.1 | 2.6 | 6.3 |
| 1221 | Above $Z$-pole | 29.8 | 2.4 | |
| | $Z, W$-pole + $\tau$ | 59.0 | | |
| 1331 | Above $Z$-pole | 29.8 | 2.4 | |
| | $Z, W$-pole + $\tau$ | 9.9 | | |
| 2211 | Above $Z$-pole | | 18.4 | |
| | $Z, W$-pole + $\tau$ | | 3.1 | |
| 3311 | Above $Z$-pole | | 18.4 | |
| | $Z, W$-pole + $\tau$ | | 2.8 | |

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
