# Peer review of "New Physics Through Flavor Tagging at FCC-ee"

_SciPost Physics, doi:SciPost Phys. 18, 152 (2025)_

## Round 1 · Referee Report · Anonymous (Referee 1) · 2025-2-10

Strengths

  • self-contained paper which guides the reader without relying to much on external knowledge
  • very clearly structed
  • as far as I can tell the results were presented exhaustively, in particular also areas where FCC-ee will not be compatible have been addessed and discussed

Weaknesses

  • minor clarifications could improve the draft

Report

The paper is an excellent read and covers the topic exhaustively. It meets the criteria of the journal and I recommend publication. Some minor annotations are provided in the attached pdf document (there were no line numbers in the draft).

Requested changes

I request no major change on the draft. A clarification on the raised points in the annotated document would be beneficial.

Attachment

Recommendation

Publish (easily meets expectations and criteria for this Journal; among top 50%)

  • validity: top
  • significance: top
  • originality: high
  • clarity: top
  • formatting: perfect
  • grammar: perfect

Author:  Alessandro Valenti  on 2025-02-24  [id 5241]

(in reply to Report 1 on 2025-02-10)
Category:
answer to question
correction

We deeply thank the referee for the positive feedback and insightful comments on our manuscript. We have revised the draft accordingly in the places we deemed appropriate. Below, we provide detailed replies to the comments, numbered according to their order in the report:

1) We believe that introducing a concrete example at this point in the paper would divert from the main narrative of the section. However, we would like to emphasize that our paper itself serves as a concrete example of corners at the TeV scale that have remained obscure even after LHC searches (e.g., four-fermion flavor-conserving operators involving second, third generation quarks).

2) We have added a footnote to clarify why we grouped up and down quarks together as $jj$.

3) Following the referee’s suggestion, we have swapped the paragraphs summarizing the contents of Sections 4 and 5.

4), 10) The normalization of hadronic and leptonic ratios follows the convention adopted at LEP and LEP-II, which we also applied to $R_t$. In particular, top quark pair production is excluded from the denominator normalization factor, as its experimental signature differs significantly from that of light quarks.

5) We clarify the advantage of directly fitting $N_{\rm{tot}}$ in the subsequent sentence: it ensures that any external uncertainty on $N_{\rm{tot}}$ does not propagate into additional uncertainty on the ratios.

6) In the preceding paragraph, above Eq. (3), we mention that we expect millions of events, justifying the use of the Gaussian approximation to the likelihood.

7) We agree with the referee that determining the background in a data-driven way would be ideal, and that further exploring this direction would be valuable. However, we have shown that an error of 1% is sufficient to leave our results on hadronic and leptonic ratios unchanged, and this level of precision already appears feasible through Monte Carlo modeling.

8), 9) The referee is correct in pointing out that the absence of correlation between relative errors is an assumption we make. The results we obtain for $R_b, R_c, R_s$, including their (small) correlation, depend on our set of assumptions, among which the one concerning the correlation of relative errors. We believe this assumption is consistent with the existing literature on the topic.

11) We consider the estimate provided in Ref. [30] for the improvement on $N_\nu$ with the best measurement to be reasonable, and we have decided to maintain it.

12) We confirm that this corresponds to the $\Delta \chi^2 = 2.3$ ellipsis.

13) We report the confidence intervals from LHC and HL-LHC as given in the external references cited in the text, and for consistency adopt the same convention in our results.

14) As explained in the text, the bounds from the Z-pole and those above the Z-pole are dominated by different observables, which depend on different combinations of Wilson coefficients. Consequently, it is reasonable to expect that the corresponding directions are not perfectly aligned, allowing their combined fit to resolve most of the correlations within each one individually. The fact that they are nearly orthogonal is a fortunate numerical coincidence.

15) We have added "current" to the text to clarify that we are comparing to current Bs meson bounds.

We hope that our revisions adequately address all concerns.

Best regards,

the authors.

---

## Round 1 · Referee Report · Anonymous (Referee 2) · 2025-4-14

Strengths

  • novel analysis for FCC-ee and FCC-hh
  • clear and exhaustive
  • pivoltal phenomenological reach

Report

This work assesses the importance of flavour tagging at FCC-ee for measuring hadronic cross-section. The results show the importance of these measurements and their impact on phenomenology. I have no reservations about publishing this paper.

Recommendation

Publish (surpasses expectations and criteria for this Journal; among top 10%)

---

## Editorial Decision

published